# SCALING SPEECH TOKENIZERS WITH DIFFUSION AUTOENCODERS

**Yuancheng Wang**[1,2]*, **Zhenyu Tang**[1], **Yun Wang**[1], **Arthur Hinsvark**[1], **Yingru Liu**[1],
**Yinghao Li**[1], **Kainan Peng**[1], **Junyi Ao**[1,2], **Mingbo Ma**[1], **Mike Seltzer**[1], **Qing He**[1], **Xubo Liu**[1]

[1] Meta Superintelligence Labs
[2] The Chinese University of Hong Kong, Shenzhen

## ABSTRACT

Speech tokenizers are foundational to speech language models, yet existing approaches face two major challenges: (1) balancing trade-offs between encoding semantics for understanding and acoustics for reconstruction, and (2) achieving low bit rates and low token rates. We propose *Speech Diffusion Tokenizer* (*SiTok*), a diffusion autoencoder that jointly learns semantic-rich representations through supervised learning and enables high-fidelity audio reconstruction with diffusion. We scale SiTok to 1.6B parameters and train it on 2 million hours of speech. Experiments show that SiTok outperforms strong baselines on understanding, reconstruction and generation tasks, at an extremely low token rate of 12.5 Hz and a bit-rate of 200 bits-per-second.

## 1 INTRODUCTION

Speech tokenizers (Parker et al., 2024; Guo et al., 2025; Mousavi et al., 2025) define the interface between continuous speech signals and discrete speech language models (Borsos et al., 2023; Nguyen et al., 2025; Défossez et al., 2024; Grattafiori et al., 2024; Zeng et al., 2024b;a; Fang et al., 2024; Wang et al., 2024; Huang et al., 2025; Ding et al., 2025; Xu et al., 2025), directly shaping how speech is perceived, modeled, and generated. As speech language models continue to scale and unify understanding and generation, the quality and structure of their discrete speech representations become a critical bottleneck.

An effective speech tokenizer is generally expected to balance three competing objectives: **(1) extreme compression** for scalable language modeling, **(2) high-fidelity reconstruction** for natural speech generation, and **(3) semantic-rich representations** for downstream speech understanding. However, existing speech tokenizers struggle to satisfy these objectives jointly, particularly in the *low token-rate regime*, where each discrete token is required to encode substantial information.

Most prior approaches address this tension through heuristic compromises rather than principled solutions. (1) To mitigate reconstruction degradation at low bitrates, many methods rely on residual vector quantization (RVQ) (Zeghidour et al., 2021; Défossez et al., 2022; Kumar et al., 2023) or increased frame rates (Xin et al., 2024; Ji et al., 2024; Ju et al., 2024; Ye et al., 2025b; Wang et al., 2025a), which directly inflate token budgets and undermine efficiency for language modeling. (2) Conversely, tokenizers optimized primarily for acoustic fidelity often neglect linguistic structure, resulting in representations that are poorly suited for speech understanding. (3) Some approaches rely on multi-stage training pipelines (Guo et al., 2024; Anastassiou et al., 2024; Du et al., 2024a; Zeng et al., 2024a; Zhang et al., 2025c; Wang et al., 2025c), where representation learning is decoupled from waveform reconstruction, requiring a separate second-stage token-to-waveform mapping and thus preventing end-to-end joint optimization.

In this work, *we aim to explore a speech tokenizer paradigm that simultaneously achieves extreme compression, high-quality reconstruction, and effective representations for speech language modeling.* Crucially, we observe that under traditional acoustic reconstruction objectives, *simply scaling training data or model size yields diminishing returns at low token rates*. This limitation suggests

---

*Work done during an internship at Meta. Email: `yuanchengwang@link.cuhk.edu.cn`

a structural bottleneck imposed by vector quantization: when trained solely with deterministic reconstruction losses, aggressive compression forces the discrete latent space to collapse uncertainty, often prioritizing low-level signal details over linguistically meaningful structure. As a result, such tokenizers yield suboptimal semantic representations for downstream speech understanding tasks such as automatic speech recognition (ASR) (Zhang et al., 2023; Défossez et al., 2024). To overcome this bottleneck, a speech tokenizer requires a generative framework that can explicitly model the uncertainty induced by aggressive quantization, rather than enforcing a deterministic inverse mapping. Diffusion models (Ho et al., 2020; Song et al., 2021; Lipman et al., 2023) offer a natural solution, as they learn to reverse a stochastic corruption process and have demonstrated strong generative capability and scalability across many domains (Rombach et al., 2022; Liu et al., 2023; Shen et al., 2024; Le et al., 2023; Polyak et al., 2024). Some prior works (Anastassiou et al., 2024; Guo et al., 2024; Du et al., 2024a;b; Zhang et al., 2025c;b) have explored diffusion-based speech tokenization, but typically adopt a two-stage design: first quantizing speech self-supervised representations (Baevski et al., 2020; Chung et al., 2021; Hsu et al., 2021; Chen et al., 2022; Chiu et al., 2022), and then training a separate diffusion model for waveform or mel-spectrogram synthesis. Such decoupled training prevents the quantizer from being optimized for reconstruction fidelity and forces the diffusion decoder to adapt to suboptimal discrete codes. In contrast, we jointly optimize quantization and reconstruction within a **diffusion autoencoder** (Rey et al., 2019; Preechakul et al., 2022), ensuring that discrete codes are both highly compressive and explicitly aligned with the generative distribution of speech.

Beyond reconstruction, we introduce **semantic regularization** to explicitly shape the discrete latent space. Speech tokenizers trained solely with reconstruction losses tend to emphasize acoustic fidelity while lacking alignment with linguistic information, which is detrimental for speech language modeling in both understanding and generation tasks. Prior works (Zhang et al., 2023; Défossez et al., 2024; Li et al., 2025a; Della Libera et al., 2025) attempt to alleviate this issue through semantic distillation, aligning latent representations with self-supervised speech features via MSE or cosine similarity losses. However, such indirect alignment does not explicitly enforce linguistic consistency. In this work, we directly impose semantic supervision on the quantized latent space by introducing an auxiliary Connectionist Temporal Classification (CTC) decoder and optimizing it with a CTC loss (Graves et al., 2006), encouraging discrete tokens to preserve semantic-rich and linguistically meaningful structure. We provide a more comprehensive review of related work on low-bitrate speech tokenizers and diffusion-based speech tokenization in the Appendix B.

Since diffusion models require iterative sampling during inference, decoding efficiency becomes a key challenge. We further investigate shortcut fine-tuning (Frans et al., 2024) and additional acceleration techniques that substantially reduce the number of diffusion steps (e.g., 2 or 4) while maintaining high reconstruction quality.

In summary, we propose the *Speech Diffusion Tokenizer* (*SiTok*), scaling it to 1.6B parameters and training it on 2 million hours of speech data. SiTok achieves strong performance under an extreme compression setting of **12.5 Hz token rate and 0.2 kbps**. We comprehensively evaluate SiTok on both speech reconstruction and diverse understanding tasks, including emotion recognition, keyword spotting, speaker verification, and automatic speech recognition. In addition, we demonstrate that the learned discrete representations can be effectively used for speech generation, enabling high-quality synthesis under the same low token-rate setting. Extensive ablation studies on codebook size, codebook dimension, and residual vector quantization (RVQ) further provide insights into the design of scalable diffusion-based speech tokenizers. Reconstruction and speech generation samples are shown in `https://sitok-demo.github.io/`.

## 2 METHOD

In this section, we introduce SiTok. We first present the speech tokenization architecture based on a diffusion autoencoder. We then describe our key design, semantic regularization. Finally, we explain how decoding can be accelerated through shortcut fine-tuning, along with additional techniques that further improve reconstruction quality. Figure 1 provides an overview of our model.

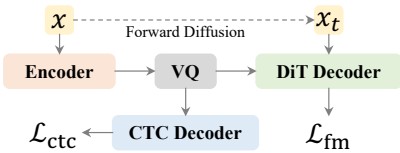

Figure 1: Overview of SiTok.

## 2.1 OVERVIEW

Speech tokenizers are generally based on autoencoders: the raw speech representation is first mapped into latent features by an encoder, then a quantizer encodes the latent features into a sequence of discrete tokens, and finally a decoder reconstructs the raw speech representation from these discrete tokens. Some speech tokenizers directly use raw waveform signals as modeling targets and rely on adversarial training to improve perceptual quality. However, we argue that this paradigm is unfavorable for scaling because: (1) directly processing the waveform sequence is inefficient due to its excessive length, which necessitates substantial up- and down-sampling and often forces prior work to cut waveforms into only a few seconds; and (2) adversarial training requires complicated loss designs and additional discriminator optimization, which tends to be unstable.

In this work, we propose a novel framework: (1) we utilize **mel-spectrograms** as both the input and the reconstruction target, leveraging a vocoder to directly synthesize the corresponding waveform; and (2) we **replace adversarial training with a diffusion autoencoder**, which facilitates more stable and scalable training. We posit that this diffusion-based framework can achieve superior compression and reconstruction. By learning to reverse the diffusion process, the model is trained to effectively capture the underlying data distribution, enabling a more robust recovery of the original signal from its quantized representation. Formally, given an input mel-spectrogram $\boldsymbol{x}$, the training process is as follows:

1. **Downsampling**: The temporal resolution of $\boldsymbol{x}$ is reduced for computational efficiency. In this work, we set the default frame rate to 12.5 Hz.
2. **Encoding**: The encoder $\mathcal{E}_\theta$ maps the downsampled spectrogram $\boldsymbol{x}$ to a sequence of latent features:
$$\boldsymbol{z} = \mathcal{E}_\theta(\boldsymbol{x}).$$
3. **Quantization**: Each feature vector in the latent sequence $\boldsymbol{z}$ is mapped to its closest entry in a discrete codebook $\boldsymbol{E} = \{\boldsymbol{e}_1, \boldsymbol{e}_2, \ldots, \boldsymbol{e}_K\}$, producing a sequence of discrete indices $\boldsymbol{q}$. This vector quantization step can be denoted as:
$$\boldsymbol{q} = \mathrm{VQ}(\boldsymbol{z}; \boldsymbol{E}).$$
4. **Diffusion Modeling**: The decoder $\mathcal{D}_\phi$ is trained to reconstruct $\boldsymbol{x}$ conditioned on the quantized representation. The discrete indices $\boldsymbol{q}$ are first mapped back to their corresponding codebook embeddings $\boldsymbol{z}_q$. Using a flow-matching objective, the decoder $\mathcal{D}_\phi$ learns to predict a velocity field that transforms a noisy sample back to the original data. The process is defined as:
$$\boldsymbol{x}_t = t\boldsymbol{x} + (1 - t)\,\boldsymbol{\epsilon}, \quad \text{where } \boldsymbol{\epsilon} \sim \mathcal{N}(\boldsymbol{0}, \boldsymbol{I}) \text{ and } t \sim U(0, 1).$$

The decoder's predicted velocity $v_\phi$ is optimized to match the true velocity $(\boldsymbol{x} - \boldsymbol{\epsilon})$:
$$v_\phi(\boldsymbol{x}_t, t, \boldsymbol{z}_q) = \mathcal{D}_\phi(\boldsymbol{x}_t, t, \boldsymbol{z}_q) \to \boldsymbol{x} - \boldsymbol{\epsilon}.$$

## 2.2 SEMANTIC REGULARIZATION

In our preliminary study, we found that relying solely on reconstruction, whether based on a diffusion loss or a regression loss, results in poor intelligibility (much higher WER) and degraded performance on downstream understanding tasks. Motivated by prior works, **we introduce an auxiliary loss to directly supervise the latent space after vector quantization.** Unlike approaches that employ representation alignment (Yu et al., 2025) or semantic distillation (Zhang et al., 2023; Défossez et al., 2024) to match the latent representations with features from self-supervised models, we directly predict the textual content through an additional lightweight decoder $\mathcal{D}_{\phi_{\mathrm{ctc}}}$ trained with a CTC loss. Some previous works like Beichuan-Audio tokenizer (Li et al., 2025b) and XY-Tokenizer (Gong et al., 2025) also incorporate ASR-based supervision to enrich semantic representations, but they still rely on an additional ASR model as a semantic encoder to extract latent features. In contrast, SiTok learns representations directly from raw speech.

The total loss $\mathcal{L}_{\mathrm{total}}$ combines three components: the diffusion reconstruction loss ($\mathcal{L}_{\mathrm{rec}}$), the semantic CTC loss ($\mathcal{L}_{\mathrm{ctc}}$), and the vector quantization loss ($\mathcal{L}_{\mathrm{vq}}$). Given the ground-truth text transcript $\boldsymbol{y}$, the objective is:
$$\mathcal{L}_{\mathrm{total}} = \underbrace{\mathbb{E}_{t,\boldsymbol{x},\boldsymbol{\epsilon}} \left[\|\mathcal{D}_\phi(\boldsymbol{x}_t, t, \boldsymbol{z}_q) - (\boldsymbol{x} - \boldsymbol{\epsilon})\|\right]}_{\text{Reconstruction Loss}} + \lambda_{\mathrm{ctc}} \underbrace{\mathrm{CTC}(\mathcal{D}_{\phi_{\mathrm{ctc}}}(\boldsymbol{z}_q), \boldsymbol{y})}_{\text{CTC Loss}} + \underbrace{\mathcal{L}_{\mathrm{vq}}}_{\text{VQ Loss}}$$

where $\boldsymbol{z}_q$ is the sequence of quantized embeddings, $\mathcal{D}_{\phi_{\text{ctc}}}$ is the auxiliary CTC decoder, and $\lambda_{\text{ctc}}$ is a balancing hyperparameter. We find that $\lambda_{\text{ctc}}$ is crucial for performance.

## 2.3 EFFICIENT DECODING

Traditional diffusion models often require multiple inference steps, which can make decoding computationally inefficient. To address this, we explore two strategies to significantly accelerate diffusion decoding: **Shortcut Fine-tuning** and **Light-weight Diffusion Head**.

**Shortcut Fine-tuning**    We explore efficient few-step decoding with fine-tuning the decoder using the shortcut model objective proposed by Frans et al. (2024). During the fine-tuning stage, we freeze the encoder and VQ modules. The fine-tuning process then updates only the decoder weights. The key idea behind shortcut models is to train a network that is conditioned not only on the time step $t$ but also on a desired step size $d$. This allows the model to learn a direct mapping from a noisy input to a significantly denoised output in a single forward pass, effectively "jumping" over many intermediate steps of a standard iterative diffusion process.

The fine-tuning employs a combined loss function that jointly optimizes a flow-matching objective and a self-consistency objective. The total loss is formulated as:

$$\mathcal{L}_{\text{S}} = \mathbb{E}_{\boldsymbol{x}_0, \boldsymbol{x}_1, t, d} \Big[ \underbrace{\|\boldsymbol{s}_\phi(\boldsymbol{x}_t, t, 0) - (\boldsymbol{x}_1 - \boldsymbol{x}_0)\|_2^2}_{\text{Flow-Matching Loss}} + \underbrace{\|\boldsymbol{s}_\phi(\boldsymbol{x}_t, t, 2d) - \boldsymbol{s}_{\text{target}}\|_2^2}_{\text{Self-Consistency Loss}} \Big]$$

where $\boldsymbol{s}_\phi$ is the shortcut model (our decoder) being trained. The target for the self-consistency loss, $\boldsymbol{s}_{\text{target}}$, is generated by the model itself using two consecutive smaller steps, with gradients detached:

$$\boldsymbol{s}_{\text{target}} = \text{stopgrad}\left( \frac{1}{2} \boldsymbol{s}_\phi(\boldsymbol{x}_t, t, d) + \frac{1}{2} \boldsymbol{s}_\phi(\boldsymbol{x}_{t+d}, t+d, d) \right)$$

and $\boldsymbol{x}_{t+d} = \boldsymbol{x}_t + \boldsymbol{s}_\phi(\boldsymbol{x}_t, t, d)d$.

The first term grounds the model's behavior for infinitesimal step sizes ($d = 0$), ensuring it matches the true data velocity. The second term enforces that a single large step of size $2d$ yields the same result as two sequential steps of size $d$. This self-consistency training enables the decoder to accurately perform large, discrete jumps along the denoising trajectory, significantly reducing inference steps while maintaining high reconstruction quality.

**Light-weight Diffusion Head**    We also explore a light-weight diffusion head that reduces the cost of iterative denoising by splitting the decoder into a main body (run once) and a small head reused across diffusion steps. This design lowers per-step computation; see Appendix C.1 for details.

## 2.4 RECONSTRUCTION REFINEMENT

To further enhance the quality of the reconstruction speech, we employ two distinct refinement strategies. The first is a **decoder finetuning**, where the encoder and VQ modules are frozen, and only the decoder is trained further. This step specializes the decoder for high-fidelity synthesis from the fixed discrete representations. The second is the introduction of **token classifier-free guidance (Token CFG)**. To enable this, we train the decoder to be conditionally dependent on the discrete tokens by randomly dropping all input tokens with a 10% probability, which forces the decoder to also learn an unconditional generation objective. During inference, this allows us to steer the decoding process by combining predictions from both conditional (with tokens) and unconditional (with dropped tokens) passes, leading to a more robust and accurate reconstruction. The efficacy of both optional refinement techniques is empirically validated in our results (Table 1).

## 3 EXPERIMENTS AND RESULTS

### 3.1 IMPLEMENTATION DETAILS

**Data and Preprocessing**    We use 2 million of in-house speech data to train our models. The dataset covers multiple languages, with English accounting for the vast majority. We do not apply additional preprocessing to the speech data, such as splitting into shorter segments; instead, we train

directly on the original utterance lengths paired with their transcripts. We use 50 Hz, 128-bin mel-spectrograms as both the input and reconstruction targets of our tokenizer, while first stacking every four consecutive frames to reduce the frame rate to 12.5 Hz for more efficient training. For waveform synthesis, we employ a Vocos-based (Siuzdak, 2024) vocoder to convert the mel spectrograms back to audio waveforms at 24K Hz.

**Model**  Our model is constructed using standard Llama-style Transformer blocks (Touvron et al., 2023; Grattafiori et al., 2024). The encoder and the auxiliary CTC decoder are composed of causal Llama decoder layers, with 16 and 4 layers, respectively. Unless otherwise specified, we set the hidden size to 1536, the intermediate size to 4096, and the number of attention heads to 16. For the VQ module, we adopt a default configuration of 32 dimensions with a codebook of 65,536 entries, updated using an exponential moving average (EMA) (van den Oord et al., 2017). The diffusion decoder is implemented by modifying the causal Llama decoder layers into a non-causal form with 16 layers. We incorporate the diffusion step embedding by replacing RMSNorm (Zhang & Sennrich, 2019) with an Adaptive RMSNorm variant. Additional architectural details are provided in Appendix A, while ablation studies on the codebook dimension, codebook size, and overall model size are presented in Section 3.4.

**Training**  We train all models for a single epoch, corresponding to approximately 450K steps. For optimization, we adopt the AdamW (Loshchilov & Hutter, 2019) optimizer with $\beta_1 = 0.9$, $\beta_2 = 0.999$, a weight decay of $0.01$, and a learning rate of $8 \times 10^{-5}$ with a warmup of 32K steps.

## 3.2 EVALUATION

We evaluate our tokenizers from the perspectives of compression, reconstruction, and speech under-standing. We also evaluate SiTok on speech generation via zero-shot TTS, with results reported in Appendix C.2.

**Compression**  We assess the efficiency of the tokenizer in terms of token rate (TPS: tokens per second), frame rate (FPS: frames per second), and bitrate (BR). These metrics directly reflect the trade-off between compression and representational capacity.

**Reconstruction**  To evaluate speech reconstruction quality, we measure intelligibility, speaker sim-ilarity, and speech quality. Intelligibility is assessed using word error rate (WER), computed with `whisper-large-v3` (Radford et al., 2023). Speaker similarity (SIM) is computed as the cosine similarity between `WavLM-TDNN` embeddings of the prompt and the generated speech (Chen et al., 2022). Speech quality is evaluated using the official UTMOS checkpoint (Saeki et al., 2022). We report these results on the SeedTTS *test-en* (Anastassiou et al., 2024) evaluation set.

**Understanding**  We evaluate the learned representations on three speech understanding tasks: emotion recognition (ER), speaker verification (SV), and keyword spotting (KS), following the setup of the DASB benchmark (Mousavi et al., 2024). Additionally, following Yang et al. (2025), we train an LLM-based ASR (LLM ASR) model with a 1B-parameter LLM backbone, which takes the dis-crete speech tokens generated by the speech tokenizer as input and autoregressively predicts the corresponding text. We also report ASR results (CTC ASR) using the direct CTC decoder of our tokenizers. The ASR evaluation is conducted on the LibriSpeech *test-clean* (Panayotov et al., 2015).

**Evaluation Baselines**  We also compare our approach with a range of open-source speech tokeniz-ers, see more details about the baselines in the following sections.

## 3.3 RESULTS AND COMPARISON

In this section, we present a comprehensive evaluation of our proposed speech tokenizer. We first report the *main results for speech reconstruction* in Section 3.3.1, where we compare against a wide range of existing tokenizers under different compression settings. We then evaluate *downstream performance* in Section 3.3.2, covering diverse understanding tasks. Beyond these comparisons, we further analyze the *effectiveness of semantic regularization* in Section 3.3.3, *the impact of scaling model size* in Section 3.3.4, and *efficient decoding strategies* in Section 3.3.5 that improve inference speed without sacrificing quality. Finally, we conduct an extensive *ablation study* 3.4 to isolate the contributions of different components, including loss design, codebook configurations, and frame rate choices, providing insights into the design principles of scalable speech tokenizers.

### 3.3.1 Main Results for Reconstruction

Table 1: Main reconstruction results. "Decoder Finetuning" indicates that the encoder and VQ are frozen while the decoder is further trained for several steps. "Token CFG" denotes the use of classifier-free guidance, more details are shown in Section 3.4. "CN" means codebook number.

| Model | FPS/TPS | CN | BR (kbps) | WER (↓) | SIM (↑) | UTMOS (↑) |
|---|---|---|---|---|---|---|
| Ground Truth | - | - | - | 2.14 | 0.730 | 3.53 |
| SpeechTokenizer (Zhang et al., 2023) | 50/100 | 2 | 1.00 | 7.98 | 0.468 | 2.47 |
| BigCodec (Xin et al., 2024) | 80/80 | 1 | 1.04 | 3.25 | 0.615 | 3.59 |
| DualCodec (Li et al., 2025a) | 12.5/75 | 6 | 0.925 | 2.63 | 0.624 | 3.78 |
| WavTokenizer (Ji et al., 2024) | 75/75 | 1 | 0.90 | 6.65 | 0.483 | 3.36 |
| Mimi (Défossez et al., 2024) | 12.5/75 | 6 | 0.825 | 4.51 | 0.527 | 3.09 |
| X-codec 2 (Ye et al., 2025b) | 50/50 | 1 | 0.80 | 2.63 | 0.620 | 3.68 |
| SemantiCodec (Liu et al., 2024) | 25/50 | 2 | 0.675 | 5.11 | 0.488 | 2.83 |
| BiCodec (Wang et al., 2025a) | 50/50 | 1 | 0.65 | 3.05 | 0.612 | 3.68 |
| Vevo Tokenizer (Zhang et al., 2025c) | 50/50 | 1 | 0.65 | 3.04 | 0.534 | 3.50 |
| StableCodec (Parker et al., 2024) | 25/25 | 1 | 0.40 | 11.1 | 0.410 | 3.87 |
| FireRedTTS Tokenizer (Guo et al., 2024) | 25/25 | 1 | 0.35 | 3.35 | 0.597 | 3.40 |
| CosyVoice Tokenizer (Du et al., 2024a) | 25/25 | 1 | 0.30 | 5.63 | 0.465 | 3.65 |
| *SiTok* (CN = 1) | 12.5/12.5 | 1 | 0.20 | 4.06 | 0.641 | 3.44 |
| *+ Decoder Finetuning* | 12.5/12.5 | 1 | 0.20 | 3.79 | 0.682 | 3.48 |
| *+ Token CFG* | 12.5/12.5 | 1 | 0.20 | 3.34 | 0.635 | 3.60 |
| *SiTok* (CN = 2) | 12.5/25 | 2 | 0.35 | 3.17 | 0.658 | 3.44 |
| *SiTok* (CN = 4) | 12.5/50 | 4 | 0.70 | 2.80 | 0.660 | 3.46 |

Table 1 presents a detailed comparison of our speech tokenizers against other baseline models on the reconstruction task. Our model demonstrates exceptional performance under a highly challenging setting. At its base configuration (single codebook), our tokenizer operates at an extremely low bitrate of 0.2 kbps and a token rate of only 12.5 Hz, which is significantly lower than all competing methods. Despite this extreme compression, it achieves highly competitive results. We also demonstrate that the model's performance can be significantly enhanced with simple yet effective strategies. Decoder finetuning boosts speaker similarity to a remarkable 0.682. Applying token classifier free guidance reduces WER to 3.34. In addition, increasing the number of codebooks via RVQ yields consistent improvements in both WER and similarity.

### 3.3.2 Downstream Understanding

**Understanding Tasks** Table 2 shows that our tokenizer significantly outperforms all baselines on several speech understanding tasks. In particular, we achieve substantial improvements on LLM-based ASR (WER 4.95), and consistently surpass all baselines on ER, SV, and KS. LLM-based ASR results for baseline models are adopted from Yang et al. (2025). "CN" means codebook number and "CS" means codebook size.

Table 2: Main results for understanding tasks.

| Model | FPS/TPS | CN/CS | BR (kbps) | CTC ASR (↓) | ASR (↓) | ER (↑) | SV (↓) | KS (↑) |
|---|---|---|---|---|---|---|---|---|
| DAC Rombach et al. (2022) | 50/150 | 3/1024 | 1.5 | - | 58.4 | 48.9 | 17.8 | 68.8 |
| EnCodec Défossez et al. (2022) | 50/150 | 3/1024 | 1.5 | - | 77.2 | 47.4 | 15.5 | 79.3 |
| Mimi Défossez et al. (2024) | 12.5/100 | 8/2048 | 1.1 | - | 23.1 | 54.3 | 19.7 | 92.2 |
| WavTokenizer Ji et al. (2024) | 40/40 | 1/4096 | 0.48 | - | 45.6 | 51.1 | 19.4 | 65.3 |
| StableCodec Parker et al. (2024) | 25/25 | 1/46656 | 0.40 | - | 28.0 | - | - | - |
| GLM4-Voice Zeng et al. (2024a) | 12.5/12.5 | 1/16384 | 0.20 | - | 16.3 | - | - | - |
| *SiTok* (CN = 1) | 12.5/12.5 | 1/65536 | 0.20 | 9.50 | 4.95 | 63.5 | 13.8 | 96.9 |
| *SiTok* (CN = 4) | 12.5/50 | 4/16384 | 0.70 | 8.30 | 4.49 | 64.4 | 8.59 | 97.7 |

### 3.3.3 Effectiveness of Semantic Regularization

Table 3 demonstrates the impact of semantic regularization on both reconstruction and understanding tasks, which is key to enhancing reconstruction quality and learning meaningful representations

for downstream understanding. Without regularization, the model shows severely degraded intelligibility (WER rising from 4.06 to 33.0) and poor downstream performance. In contrast, applying CTC-based regularization substantially reduces WER, improves similarity and speech quality, and boosts all understanding tasks. This highlights that CTC supervision anchors the quantized latent space to linguistic meaning, ensuring tokens are both acoustically faithful and semantically informative, especially for low-rate tokenizers.

Table 3: Effectiveness of semantic regularization on reconstruction and understanding.

| CTC Reg. | FPS/TPS | Reconstruction | | | Understanding | | | | |
|---|---|---|---|---|---|---|---|---|---|
| | | WER ($\downarrow$) | SIM ($\uparrow$) | UTMOS ($\uparrow$) | CTC ASR ($\downarrow$) | ASR ($\downarrow$) | ER ($\uparrow$) | SV ($\downarrow$) | KS ($\uparrow$) |
| Yes | 12.5/12.5 | 4.06 | 0.641 | 3.44 | 9.50 | 4.95 | 63.5 | 13.8 | 96.9 |
| | 12.5/25 | 3.17 | 0.658 | 3.44 | 8.64 | 4.72 | 61.7 | 11.1 | 97.8 |
| | 12.5/50 | 2.80 | 0.660 | 3.46 | 8.30 | 4.49 | 64.4 | 8.59 | 97.7 |
| No | 12.5/12.5 | 33.0 | 0.495 | 2.68 | - | 29.4 | 57.9 | 18.9 | 86.1 |
| | 12.5/25 | 10.1 | 0.598 | 2.99 | - | 9.53 | 55.3 | 15.5 | 92.7 |
| | 12.5/50 | 5.17 | 0.611 | 2.84 | - | 7.27 | 60.4 | 13.5 | 92.8 |

> **Observation:** Applying semantic regularization to the quantized latent space is crucial for both reconstruction and representation learning, particularly when operating at low token rates.

### 3.3.4 EFFECTIVENESS OF MODEL SCALING

Our model scaling experiments from the 0.63B "S" model to the 1.61B "XL" model reveal a clear trade-off between reconstruction fidelity and downstream task performance, as shown in Table 4. While larger models consistently yield better reconstruction quality, with the XL model achieving the best WER, SIM and UTMOS, performance on understanding tasks peaks with the 1.12B "L" model, which delivers superior results in ASR. The fact that the largest model does not uniformly outperform its smaller counterparts, and even shows degradation in tasks like SV, suggests that excessive model capacity may prioritize fine-grained acoustic details over the abstract, discriminative features crucial for understanding. Therefore, we identify the "L" model as the optimal configuration, providing the most effective balance between high-quality synthesis and robust generalization. Further exploration of architectural designs is left for future work.

Table 4: Results for model size scaling. We vary the number of encoder and decoder layers while keeping the CTC layers fixed to evaluate the impact on both reconstruction and understanding tasks.

| Size | Enc. | Dec. | Params (B) | Reconstruction | | | Understanding | | | | |
|---|---|---|---|---|---|---|---|---|---|---|---|
| | | | | WER ($\downarrow$) | SIM ($\uparrow$) | UTMOS ($\uparrow$) | CTC ASR ($\downarrow$) | ASR ($\downarrow$) | ER ($\uparrow$) | SV ($\downarrow$) | KS ($\uparrow$) |
| S | 8 | 8 | 0.63 | 4.18 | 0.608 | 3.43 | 11.2 | 5.24 | 60.8 | 13.7 | 96.9 |
| B | 12 | 12 | 0.88 | 4.01 | 0.634 | 3.46 | 9.78 | 5.19 | 62.5 | 13.8 | 96.9 |
| L | 16 | 16 | 1.12 | 4.06 | 0.641 | 3.44 | 9.50 | 4.95 | 63.5 | 13.8 | 96.9 |
| XL | 24 | 24 | 1.61 | 3.84 | 0.649 | 3.51 | 9.62 | 5.07 | 63.5 | 14.7 | 97.3 |

### 3.3.5 EFFICIENT DECODING

**Shortcut Fine-Tuning** Table 2 shows that directly reducing the number of inference steps leads to a clear degradation in intelligibility and audio quality. In contrast, shortcut fine-tuning substantially alleviates this issue, achieving much lower WER and higher speaker similarity even with very small numbers of diffusion steps. Moreover, this improvement also translates into a significant reduction in real-time factor (RTF): the model runs at **0.041**, **0.024**, and **0.013** RTF for **16**, **8**, and **4** diffusion steps, respectively. These results demonstrate that shortcut fine-tuning effectively adapts the model to faster sampling schedules while preserving reconstruction fidelity and enabling highly efficient inference.

> **Observation:** Shortcut fine-tuning enables efficient low-step inference, retaining high intelligibility and similarity while substantially accelerating decoding.

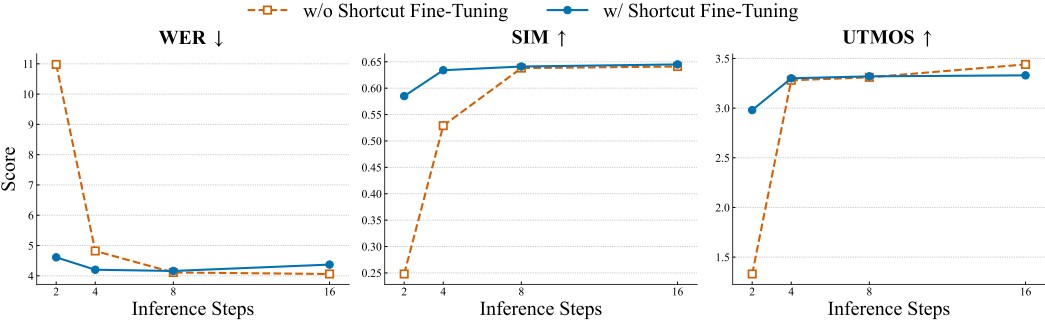

Figure 2: Impact of shortcut fine-tuning on different inference steps. We report WER, SIM, and UTMOS. Shortcut fine-tuning achieves consistently better intelligibility and similarity, especially at small step numbers.

**Light-Weight Diffusion Head**    We also explore using light-weight diffusion heads to accelerate diffusion inference. We provide the results in Appendix C.1.

### 3.3.6    SɪTᴏᴋ ꜰᴏʀ Sᴘᴇᴇᴄʜ Gᴇɴᴇʀᴀᴛɪᴏɴ

Beyond speech understanding, an important use case of a speech tokenizer is to serve as the modeling target for speech generation. To this end, we further evaluate SiTok on the zero-shot text-to-speech (TTS) task, assessing whether the learned discrete representations can effectively support high-quality speech synthesis. Experimental results show that SiTok is not only effective as a representation for speech understanding, but also enables high-quality and efficient speech generation. In particular, the extremely low token rate of SiTok significantly reduces the sequence length required for generation, leading to faster inference while maintaining strong perceptual quality. These results suggest that SiTok provides a unified representation that is well suited for both speech understanding and speech generation. Detailed experimental results for zero-shot TTS are reported in Appendix C.2.

### 3.4    Aʙʟᴀᴛɪᴏɴ Sᴛᴜᴅʏ

To better understand the contributions of different design choices in our tokenizer, we conduct a series of ablation studies, the results are shown in Table 5. We systematically examine training objectives, regularization strategies, refinement mechanisms, codebook configurations, and frame rates. These analyses not only validate the effectiveness of our proposed components but also provide insights into the trade-offs between efficiency, reconstruction quality, and downstream performance.

**Diffusion vs. Regression**    We investigate the choice of the reconstruction objective by comparing our proposed diffusion-based objective (D) against a conventional regression-based objective (R), such as an L1 loss on the mel-spectrogram. As shown in Table 5, our baseline model trained with the diffusion loss significantly outperforms the regression-based counterpart across all key metrics. Specifically, the diffusion model achieves a substantially lower WER (4.06 vs. 4.66), higher speaker similarity (0.641 vs. 0.587), and better downstream ASR performance (4.95 vs. 6.06). To further explore if a diffusion decoder could salvage a regression-trained model, we conducted an experiment where only the decoder was fine-tuned with the diffusion objective on top of a regression-pretrained model (R + D). Therefore, by design, the understanding metrics for (R) and (R + D) are identical, while reconstruction metrics (WER/SIM/UTMOS) change. While this modestly improved speaker similarity, **it failed to match the performance of the end-to-end diffusion model** and even worsened the WER to 5.73. This indicates that the representations learned under the diffusion objective are inherently superior for both high-fidelity reconstruction and downstream task transferability.

> **Observation:** Adopting a diffusion-based objective is critical for learning high-quality representations, significantly outperforming a standard regression objective in both reconstruction fidelity and downstream task performance.

Table 5: Ablation study.

| | Loss | CTC W. | CS | CN | CD | Tok. CFG | FPS | Reconstruction | | | CTC ASR (↓) | Understanding | | | |
|---|---|---|---|---|---|---|---|---|---|---|---|---|---|---|---|
| | | | | | | | | WER (↓) | SIM (↑) | UTMOS (↑) | | ASR (↓) | ER (↑) | SV (↓) | KS (↑) |
| | D | 0.1 | $2^{16}$ | 1 | 32 | ✗ | 12.5 | 4.06 | 0.641 | 3.44 | 9.50 | 4.95 | 63.5 | 13.8 | 96.9 |
| **Loss** | R | 0.1 | $2^{16}$ | 1 | 32 | ✗ | 12.5 | 4.66 | 0.587 | 3.28 | 12.2 | 6.06 | 63.3 | 13.6 | 95.2 |
| | R + D | 0.1 | $2^{16}$ | 1 | 32 | ✗ | 12.5 | 5.73 | 0.634 | 3.35 | 12.2 | 6.06 | 63.3 | 13.6 | 95.2 |
| **CTC W.** | D | 0 | $2^{16}$ | 1 | 32 | ✗ | 12.5 | 33.0 | 0.495 | 2.68 | - | 29.4 | 57.9 | 18.9 | 86.1 |
| | D | 0.02 | $2^{16}$ | 1 | 32 | ✗ | 12.5 | 5.05 | 0.607 | 3.44 | 12.2 | 7.41 | 58.3 | 13.5 | 97.2 |
| | D | 0.5 | $2^{16}$ | 1 | 32 | ✗ | 12.5 | 8.81 | 0.614 | 3.20 | 11.0 | 7.87 | 64.2 | 16.6 | 91.2 |
| | D | 1 | $2^{16}$ | 1 | 32 | ✗ | 12.5 | 10.1 | 0.585 | 3.38 | 10.5 | 8.90 | 62.1 | 13.9 | 96.8 |
| **CS** | D | 0.1 | $2^{13}$ | 1 | 32 | ✗ | 12.5 | 5.48 | 0.640 | 3.39 | 11.7 | 5.72 | 55.7 | 16.3 | 95.7 |
| | D | 0.1 | $2^{14}$ | 1 | 32 | ✗ | 12.5 | 4.30 | 0.641 | 3.33 | 11.5 | 5.40 | 58.7 | 16.0 | 96.4 |
| | D | 0.1 | $2^{15}$ | 1 | 32 | ✗ | 12.5 | 4.26 | 0.648 | 3.43 | 10.5 | 5.33 | 61.2 | 15.0 | 96.4 |
| | D | 0.1 | $2^{17}$ | 1 | 32 | ✗ | 12.5 | 3.94 | 0.651 | 3.39 | 10.6 | 5.09 | 60.6 | 13.7 | 97.3 |
| **CN** | D | 0.1 | $2^{14}$ | 2 | 32 | ✗ | 12.5 | 3.17 | 0.658 | 3.44 | 8.64 | 4.72 | 61.7 | 11.1 | 97.8 |
| | D | 0.1 | $2^{14}$ | 4 | 32 | ✗ | 12.5 | 2.80 | 0.660 | 3.46 | 8.30 | 4.49 | 64.4 | 8.59 | 97.7 |
| | D | 0.1 | $2^{14}$ | 8 | 32 | ✗ | 12.5 | 2.50 | 0.645 | 3.30 | 8.42 | 4.68 | 60.0 | 7.53 | 98.2 |
| **CD** | D | 0.1 | $2^{14}$ | 1 | 64 | ✗ | 12.5 | 4.08 | 0.642 | 3.46 | 9.58 | 5.27 | 61.7 | 14.4 | 97.3 |
| | D | 0.1 | $2^{14}$ | 1 | 128 | ✗ | 12.5 | 3.85 | 0.642 | 3.36 | 9.14 | 4.59 | 59.7 | 14.3 | 97.3 |
| | D | 0.1 | $2^{14}$ | 1 | 256 | ✗ | 12.5 | 5.04 | 0.641 | 3.30 | 10.9 | 5.54 | 64.1 | 16.5 | 95.9 |
| **Tok. CFG** | D | 0.1 | $2^{16}$ | 1 | 32 | ✓ | 12.5 | 3.34 | 0.635 | 3.60 | 9.54 | 4.89 | 62.4 | 14.0 | 96.8 |
| | D | 0.1 | $2^{14}$ | 4 | 32 | ✓ | 12.5 | 2.56 | 0.645 | 3.58 | 8.34 | 4.67 | 61.7 | 9.11 | 98.2 |
| **FPS** | D | 0.1 | $2^{16}$ | 1 | 32 | ✗ | 6.25 | 23.0 | 0.428 | 3.10 | 15.8 | 12.7 | 52.6 | 20.7 | 88.5 |
| | D | 0.1 | $2^{16}$ | 1 | 32 | ✗ | 25 | 3.05 | 0.688 | 3.72 | 9.19 | 4.45 | 63.5 | 7.28 | 97.8 |

**CTC Loss Weight** We analyze the impact of the CTC loss weight, which serves as a semantic regularizer. The results clearly demonstrate that this regularization is indispensable. Setting the weight to 0 leads to a catastrophic performance collapse, with the WER soaring to 33.0 and the downstream ASR error rate to 29.4, confirming that the model fails to learn any meaningful representations without textual supervision. Conversely, an excessively high weight (e.g., 0.5 or 1.0) also degrades performance across both reconstruction and understanding tasks, likely by forcing the model to discard too much acoustic detail in favor of semantic content. Our experiments identify a weight of 0.1 as providing the optimal balance between enforcing semantic consistency and preserving high-fidelity audio reconstruction.

> **Observation:** The CTC loss weight is a critical hyperparameter; too low a value fails to enforce semantic consistency, while too high a value impairs reconstruction fidelity.

**Reconstruction Refinement** As shown in our main reconstruction results (Table 1), both decoder finetuning and token classifier-free guidance (CFG) serve as powerful techniques to enhance reconstruction quality. Decoder finetuning, a training-time strategy, specializes the synthesis module on the fixed representations, leading to significant improvements in both intelligibility and particularly speaker similarity. Separately, applying token CFG at inference time provides a complementary approach to boost fidelity. Our ablation study demonstrates that CFG consistently and substantially reduces WER, achieving a low of 2.56 in our CD = 4 configuration, while also improving perceptual quality (UTMOS). These findings indicate that both training-time adaptation and inference-time guidance are highly effective strategies for refining the final output.

**Codebook Size** We investigate the effect of the codebook size (CS), which controls the vocabulary of discrete tokens. We find enlarging the codebook from $2^{13}$ to $2^{17}$ consistently improves reconstruction quality, evidenced by a steady decrease in WER from 5.48 to 3.94. Downstream task performance, particularly ASR, also benefits and reaches an optimum at a codebook size of $2^{16}$, suggesting this size offers the best balance between representational power and generalization.

**Codebook Number** We also evaluate increasing the number of codebooks (CN) using RVQ, which directly increases the bitrate. We find that scaling CN from 1 to 8 yields substantial and consistent improvements across most metrics. The reconstruction WER drops dramatically from 4.30 to 2.50, while downstream performance on tasks like ASR and SV is also significantly boosted. This demonstrates an effective trade-off between compression and fidelity within our framework.

> **Observation:** The vector quantizer's design offers a flexible trade-off between quality and complexity. Increasing the number of codebooks provides a direct path to higher fidelity and better downstream performance at the cost of bitrate.

**Frame Rate vs. Performance Trade-off** We also investigate two alternative frame-rate settings: 6.25 Hz and 25 Hz. We find that reducing the frame rate to 6.25 Hz significantly degrades both reconstruction and downstream task performance, while increasing it to 25 Hz improves performance but doubles the frame rate. Therefore, we adopt 12.5 Hz as the default setting to balance efficiency and performance.

## 4 CONCLUSION

In this work, we propose *SiTok*, a diffusion autoencoder–based speech tokenizer that enables end-to-end joint modeling of reconstruction and quantization for improved acoustic fidelity. We further introduce semantic regularization to learn effective, semantically rich representations for speech understanding, and explore shortcut fine-tuning techniques to significantly accelerate diffusion decoding. Extensive experiments demonstrate that SiTok achieves strong performance on both speech reconstruction and diverse speech understanding tasks. In addition, we conduct extensive ablation studies, providing insights into the key design choices.

## ETHICS STATEMENT

This work studies speech tokenizers with diffusion autoencoders. Our models are designed for academic research and downstream tasks such as speech understanding and speech language modeling. While tokenizers themselves are neutral, we acknowledge potential misuse in downstream systems (e.g., generating synthetic speech for impersonation) and encourage responsible and ethical use of our models.

## REPRODUCIBILITY STATEMENT

To ensure reproducibility, we provide detailed descriptions of model architectures, training settings, and evaluation protocols in the main paper and Appendix.

## USE OF LLMS

LLMs were employed for auxiliary purposes in this work, such as grammar checking, polishing the manuscript. However, all technical contributions, model implementations, and experimental analyses were conducted by the authors. We acknowledge the use of LLMs where appropriate and ensure that their involvement does not compromise the originality of the work.

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

## A  MODEL ARCHITECTURE

**Model**  The tokenizer's architecture consists of a causal encoder, a vector quantizer (VQ), a diffusion decoder, and a causal auxiliary CTC decoder. For the causal encoder and the causal auxiliary CTC decoder, we utilize standard Llama-style Transformer blocks (Touvron et al., 2023; Grattafiori et al., 2024), incorporating RoPE positional encoding (Su et al., 2024) and the SiLU (Elfwing et al., 2018) activation function. The encoder is specifically implemented with 16 causal Llama decoder layers, and the auxiliary CTC decoder with 4 such layers. A consistent configuration, unless otherwise specified, applies across these components (and the diffusion decoder): a hidden size of 1536, an intermediate size of 4096, and 16 attention heads. The VQ module employs a default configuration of 32 dimensions, featuring a codebook of 65,536 entries, with updates managed via an exponential moving average (EMA) (van den Oord et al., 2017). The diffusion decoder is realized using 16 layers, adapted from the causal Llama decoder structure into a non-causal form. Diffusion step embedding is incorporated by substituting RMSNorm (Zhang & Sennrich, 2019) with an Adaptive RMSNorm variant. To study the effect of model capacity, we scale the number of encoder and diffusion decoder layers while keeping other architectural settings fixed. We experiment with four configurations: **S** (8 encoder / 8 decoder layers, 0.63B parameters), **B** (12/12, 0.88B), **L** (16/16, 1.12B), and **XL** (24/24, 1.61B). Unless otherwise specified, we adopt the **L** configuration as our default setting.

## B  RELATED WORK

**Discrete Speech Tokenizer**  Speech tokenizers are foundational components for speech language models. Early approaches (Zeghidour et al., 2021; Défossez et al., 2022; Kumar et al., 2023) primarily focused on audio compression, relying on residual vector quantization (RVQ) (Zeghidour et al., 2021; Lee et al., 2022) and operating at high frame rates and bitrates, which are suboptimal for language modeling. More recent work has shifted toward tokenizers specifically designed for language modeling, emphasizing low frame rates (Défossez et al., 2024; Li et al., 2025a; Della Libera et al., 2025), semantically rich representations (Zhang et al., 2023; Défossez et al., 2024; Li et al., 2025a; Wang et al., 2025a; Ye et al., 2025a;b; Liu et al., 2024; Guo et al., 2024; Du et al., 2024b; Zhang et al., 2025c; Jiang et al., 2025; Bie et al., 2025), and simplified single-layer codebooks (Parker et al., 2024; Xin et al., 2024; Ji et al., 2024). Nonetheless, many of these tokenizers still struggle to achieve even greater compression rates, for instance, 12.5 Hz with a single codebook. While TaDi-Codec (Wang et al., 2025b) successfully reduced the frame rate to 6.25 Hz by leveraging a diffusion autoencoder and incorporating text into its decoder, this text-aware design inherently restricts its applicability, rendering it unsuitable for speech understanding tasks. In this work, our goal is to design a tokenizer which can jointly achieve a sufficient compression rate for efficient language modeling, preserve high-quality audio reconstruction, and learn effective, semantic-rich representations for understanding speech.

**Diffusion-Based Speech Tokenizers**  Diffusion-based approaches (Ho et al., 2020; Song et al., 2021) have emerged as a promising direction, showing strong scalability and robustness at low token rates. Yet, most existing methods still adopt a two-stage design: tokens are first extracted from self-supervised speech models (Baevski et al., 2020; Chung et al., 2021; Hsu et al., 2021; Chen et al., 2022; Chiu et al., 2022; Radford et al., 2023), and only then are waveforms reconstructed through diffusion. This separation limits joint optimization, as the quantizer is not trained end-to-end with the decoder. Recent efforts (Welker et al., 2025; Yang et al., 2024b) apply diffusion to improve de-tokenization fidelity, but remain constrained to relatively high token rates and still relay on two-stage modeling. Pushing diffusion-based tokenizers to ultra-low bitrates (*e.g.*, below 0.2 kbps or 20 tokens/s) in a compact, language-model-friendly framework therefore remains an open and critical challenge. In this work, we address this challenge with *SiTok*, a diffusion-based speech tokenizer that unifies vector quantization and reconstruction modeling in an end-to-end framework, while introducing semantic regularization to ensure the learned codes are both highly compressive and semantically rich for speech language modeling.

# C    MORE EXPERIMENTS AND RESULTS

## C.1    LIGHT-WEIGHT DIFFUSION HEAD

A primary cause of inefficiency in diffusion inference is the need to execute a forward pass through the entire model at each denoising step. Our preliminary experiments revealed that for our autoencoder-based tokenizer, using a simple regression loss (e.g., L1 or L2) alone can reconstruct speech with acceptable intelligibility, albeit with poor perceptual quality. This indicates that the main body of the decoder is capable of generating the fundamental structure of the speech, while the iterative diffusion process primarily serves to refine its quality and detail.

Based on this insight, we propose partitioning the decoder $\mathcal{D}_\phi$ to decouple the main structure generation from the iterative refinement. We divide the decoder into two components: a substantial main body, $\mathcal{D}_{\phi_{\text{main}}}$, and a smaller **Light-weight Diffusion Head**, $\mathcal{D}_{\phi_{\text{head}}}$. The main body consists of the initial, deeper transformer blocks, while the head is composed of the final few blocks.

During the decoding process, the quantized embedding sequence $z_q$ is first passed through the main body $\mathcal{D}_{\phi_{\text{main}}}$ only once to produce a **base representation** $h_{\text{base}}$:

$$h_{\text{base}} = \mathcal{D}_{\phi_{\text{main}}}(z_q).$$

This base representation then provides the foundational conditioning, which is subsequently refined by the diffusion head into the final spectrogram. The iterative denoising process is performed exclusively by the light-weight head $\mathcal{D}_{\phi_{\text{head}}}$, which takes the noisy spectrogram $x_t$ and the base representation $h_{\text{base}}$ as conditioning to predict the velocity:

$$v_\phi(x_t, t, h_{\text{base}}) = \mathcal{D}_{\phi_{\text{head}}}(x_t, t, h_{\text{base}}).$$

This architectural modification significantly reduces the computational overhead per inference step, as the majority of the decoder's parameters in $\mathcal{D}_{\phi_{\text{main}}}$ are utilized in just a single forward pass. This approach allows for rapid inference while retaining the high-quality synthesis capabilities of the diffusion model.

In this work, we apply the proposed light-weight diffusion head to our base model architecture with 16 transformer decoder layers. Specifically, the first 12 layers (3/4 of the decoder) are used as the main body $\mathcal{D}_{\phi_{\text{main}}}$, while the last 4 layers serve as the diffusion head $\mathcal{D}_{\phi_{\text{head}}}$. During inference, the base representation $h_{\text{base}}$ is computed once by the main body, and only the light-weight head is executed iteratively across diffusion steps. With 16 diffusion steps as the default setting, the theoretical speedup approaches a $4\times$ reduction in per-step computation compared to applying all 16 layers at every denoising step.

As shown in Table 6, this architectural modification yields nearly identical reconstruction performance to the full model. The light-weight head maintains comparable WER and perceptual metrics (SIM and UTMOS), while significantly reducing the computational cost. This demonstrates that most of the heavy-lifting for content and structure generation is handled by the main body, and the lightweight head suffices to refine acoustic detail during diffusion inference.

Table 6: Ablation study of the light-weight diffusion head.

| Model | Reconstruction | | |
|---|---|---|---|
| | **WER** | **SIM** | **UTMOS** |
| Base | 4.06 | 0.641 | 3.44 |
| w. light head | 3.97 | 0.610 | 3.46 |

## C.2    ZERO-SHOT TTS WITH SITOK

we also evaluate SiTok in a downstream speech generation task **zero-shot text-to-speech (TTS)** to further demonstrate the effectiveness and efficiency of SiTok for speech language modeling. This experiment verifies that the discrete representations learned by SiTok are not only suitable for reconstruction and understanding, but also serve as a strong generation interface for speech language models.

Table 7: Zero-shot TTS results comparing SiTok-AR-TTS with representative AR-based TTS systems. FPS/TPS follow the definition in Section 3.2. RTF is measured on a single A100 GPU.

| Model | FPS/TPS | WER ($\downarrow$) | SIM ($\uparrow$) | RTF ($\downarrow$) |
|---|---|---|---|---|
| CosyVoice 2 (Du et al., 2024b) | 25/25 | 2.89 | 0.66 | 0.455 |
| SparkTTS (Wang et al., 2025a) | 12.5/12.5 | 2.50 | 0.57 | 0.601 |
| Llasa (Ye et al., 2025b) | 50/50 | 3.94 | 0.58 | 0.422 |
| *SiTok-AR-TTS* | 12.5/12.5 | 2.46 | 0.64 | 0.234 |

We build a 0.5B-parameter LLM-based TTS model (denoted as *SiTok-AR-TTS*), initialized from `Qwen2.5-0.5B` (Yang et al., 2024a), which autoregressively predicts SiTok tokens from text. The model is trained on 100K hours of the Emilia (He et al., 2024) dataset under a standard AR-TTS training recipe. During inference, the predicted discrete token sequence is decoded by our diffusion decoder to obtain mel-spectrograms, we use a default decoding step of 16 in this experiment.

We follow the evaluation protocol in Section 3.2 and report WER and SIM on the SeedTTS *test-en* set. To further assess practical efficiency, we also report the **real-time factor (RTF)** measured on a single NVIDIA A100 GPU, averaging 10 runs of synthesizing a 10-second utterance. Results are summarized in Table 7. We use some strong AR-based TTS models as baselines.

The results show that SiTok-AR-TTS achieves competitive or superior intelligibility and speaker similarity compared to strong baselines, while operating at a substantially lower inference cost. Interestingly, the WER obtained by SiTok-AR-TTS is even lower than the reconstruction WER of SiTok itself. This trend is consistent with some recent zero-shot TTS systems Du et al. (2024a); Guo et al. (2024); Zhang et al. (2025a), where the generated speech tokens directly conditioned on the text.

Another key observation is the considerable efficiency gain. Because SiTok operates at only 12.5 Hz, the autoregressive text-to-token decoding runs on a sequence 2 to 4$\times$ shorter than those used by conventional neural codecs operating at 25 to 50 Hz. This reduction directly translates into faster inference for speech generation, and results in a significantly lower RTF of 0.234, making SiTok particularly attractive for large-scale TTS or speech generation systems. Overall, these findings demonstrate that SiTok serves as a highly effective interface for speech generation: its discrete representations not only support high-quality reconstruction and strong downstream understanding, but also enable efficient, high-fidelity TTS within a unified speech tokenization framework.

> **Observation:** SiTok provides strong zero-shot TTS performance with high intelligibility and similarity, while its extremely low token rate enables substantially faster inference compared to existing AR-based TTS systems.

## C.3 COMPARISON WITH ALTERNATIVE QUANTIZATION METHODS

SiTok is not tied to a specific quantization design. In principle, any quantization method, such as Finite Scalar Quantization (FSQ) (Mentzer et al., 2023) and Binary Spherical Quantization (BSQ) (Zhao et al., 2024), can be integrated into our diffusion autoencoder. In this section, we compare our standard VQ module with a representative alternative, Fixed-Scalar Quantization (FSQ). We follow a commonly used FSQ configuration with per-dimension cardinalities $[2, 2, 2, 2, 2, 2, 2, 2, 2, 2, 2, 2, 2, 2, 2]$, whose Cartesian product yields a codebook size of $2^{16} = 65536$, identical to the codebook size employed by our VQ setup.

Table 8 summarizes the results. Despite having the same codebook size, standard VQ achieves stronger performance across reconstruction quality and downstream understanding tasks. We attribute this performance gap to two properties of our large-scale training regime. First, the learnable embedding vectors in VQ provide greater representational flexibility than the fixed scalar partitions of FSQ, enabling richer modeling of fine-grained acoustic and semantic structure. Second, with large batch sizes, EMA updates, and diffusion-based optimization, SiTok exhibits stable codebook utilization exceeding 95% throughout training, meaning that FSQ's typical advantage, improved

quantization stability, offers limited benefit in our setting. As a result, VQ emerges as the more expressive and empirically effective choice, though our results confirm that SiTok remains compatible with a broad family of quantization techniques.

Table 8: Comparison of VQ and FSQ within SiTok.

| Model | WER ($\downarrow$) | SIM ($\uparrow$) | UTMOS ($\uparrow$) | CTC ASR ($\downarrow$) | ASR ($\downarrow$) | ER ($\uparrow$) | SV ($\downarrow$) | KS ($\uparrow$) |
|---|---|---|---|---|---|---|---|---|
| *SiTok* (VQ) | 4.06 | 0.641 | 3.44 | 9.50 | 4.95 | 63.5 | 13.8 | 96.9 |
| *SiTok* (FSQ) | 5.23 | 0.629 | 3.44 | 10.02 | 5.33 | 62.0 | 14.1 | 96.9 |

## D  REPRODUCIBILITY STATEMENT

To support reproducibility and facilitate future research, we provide comprehensive implementation details of SiTok in this appendix. Specifically, we include (1) detailed architectural specifications (Appendix A) and pseudo-code for the SiTok model (Appendix D.1), (2) pseudo-code outlining the core end-to-end training loop of the diffusion autoencoder (Appendix D.2), and (3) additional information regarding training hyperparameters, data preprocessing, and other implementation considerations (Appendix D.3).

We also confirm that we will release the full inference code and pretrained model checkpoints (on public, research-only datasets) to the research community upon publication, enabling researchers to reproduce our results and further build upon SiTok.

### D.1  PSEUDO-CODE FOR SITOK

Listing 1: Pseudo-code for SiTok.

```python
class SiTok:
    def __init__(
        self,
        in_dim=128,
        hidden_size=1536,
        intermediate_size=4096,
        encoder_layers=16,
        decoder_layers=16,
        ctc_decoder_layers=4,
        num_heads=16,
        vq_emb_dim=16,
        downsample_factor=4,
        vocab_size=32100,
    ):
        # temporal stacking (reduce FPS to 12.5 Hz)
        self.stack_in  = StackIn(downsample_factor)
        self.stack_out = StackOut(downsample_factor)

        # transformer encoder (Llama-style causal model)
        self.encoder = LlamaModel(
            hidden_size, intermediate_size,
            encoder_layers, num_heads,
            in_dim * downsample_factor)

        # vector quantizer (Binary Spherical Quantization)
        self.vq_in  = Linear(hidden_size, vq_emb_dim)
        self.vq     = BinarySphericalQuantizer(vq_emb_dim)
        self.vq_out = Linear(vq_emb_dim, hidden_size)

        # diffusion decoder (DiT-style transformer)
        self.decoder = DiT(
```

```python
            hidden_size, intermediate_size,
            decoder_layers, num_heads,
            use_cond=True, use_diff_step=True)

        # CTC semantic decoder (Llama-style causal model)
        self.ctc_decoder = LlamaModel(
            hidden_size, intermediate_size,
            ctc_decoder_layers, num_heads,
            vocab_size)

    # -------------------------------------------------------------
    # forward: training-time outputs for loss computation
    # -------------------------------------------------------------
    def forward(self, x, x_mask):
        """SiTok forward for training losses."""

        # 1) stack + encode to continuous latents
        h = self.stack_in(x)
        h = self.encoder(h, x_mask)

        # 2) vector quantization to discrete speech tokens
        z   = self.vq_in(h)
        z_q, vq_info = self.vq(z)
        cond = self.vq_out(z_q)

        # 3) forward diffusion (flow matching)
        t   = sample_uniform()              # t ~ U(0, 1)
        eps = randn_like(x)                 # eps ~ N(0, I)
        x_t = self.forward_diffuse(x, eps, t)    # noisy mel
        x_t = self.stack_in(x_t)

        # 4) diffusion decoder predicts flow / velocity
        flow_pred = self.decoder(x_t, t, cond)

        # 5) CTC semantic logits (for semantic regularization)
        ctc_logits = self.ctc_decoder(cond)

        return {
            "x": x,                          # GT mel
            "noise": eps,                    # diffusion noise target
            "flow_pred": flow_pred,          # for flow-matching loss
            "ctc_logits": ctc_logits,        # for CTC loss
            "vq_loss": vq_info["commit"],    # VQ commitment loss
        }

    # -------------------------------------------------------------
    # forward diffusion (flow matching target)
    # -------------------------------------------------------------
    def forward_diffuse(self, x, eps, t):
        """Apply forward diffusion to obtain a noisy sample x_t."""
        # x_t = (1 - alpha(t)) * eps + alpha(t) * x
        # In practice alpha(t) implements the flow-matching schedule.
        x_t = (1 - t) * eps + t * x
        return x_t

    # -------------------------------------------------------------
    # inference helpers
    # -------------------------------------------------------------
    def encode(self, x, mask):
        """Encode mel into quantized VQ embeddings / indices."""
        h = self.stack_in(x)
        h = self.encoder(h, mask)
        z = self.vq_in(h)
```

```python
        z_q, indices = self.vq(z)
        return z_q, indices

    def decode(self, z_q, prompt=None, steps=N):
        """Reverse diffusion to generate mel from VQ embeddings."""
        cond = self.vq_out(z_q)
        mel  = diffusion_reverse(self.decoder, cond, prompt, steps)
        return self.stack_out(mel)
```

## D.2  PSEUDO-CODE FOR TRAINING LOOP

Listing 2: Pseudo-Code for Training Loop of SiTok.

```python
for batch in dataloader:

    # ------------------------------------------------------------
    # 1) prepare mel features and masks
    # ------------------------------------------------------------
    x        = mel_extractor(batch.speech)     # [B, T, d]
    x_mask   = batch.speech_mask
    text_ids    = batch.text_ids               # semantic supervision
    text_mask   = batch.text_mask

    # ------------------------------------------------------------
    # 2) forward pass through SiTok
    # ------------------------------------------------------------
    out = sitok.forward(x, x_mask)
    x_gt        = out["x"]
    noise       = out["noise"]
    flow_pred   = out["flow_pred"]
    ctc_logits  = out["ctc_logits"]
    vq_loss     = out["vq_loss"]

    # ------------------------------------------------------------
    # 3) diffusion (flow-matching) loss
    # ------------------------------------------------------------
    # target velocity v* = x - eps
    flow_gt = x_gt - noise
    diff_loss = L1(flow_pred, flow_gt)

    # ------------------------------------------------------------
    # 4) CTC semantic loss
    # ------------------------------------------------------------
    ctc_loss = CTC_Loss(ctc_logits, text_ids, text_mask)

    # ------------------------------------------------------------
    # 5) total loss
    # ------------------------------------------------------------
    total_loss = diff_loss + vq_loss + lambda_ctc * ctc_loss

    # ------------------------------------------------------------
    # 6) optimization
    # ------------------------------------------------------------
    optimizer.zero_grad()
    total_loss.backward()
    clip_gradients(sitok.parameters(), max_norm=0.5)
    optimizer.step()
```

### D.3 More Implementation Details

**Data and Preprocessing**  We use 2 million hours of in-house data to train our models. The dataset covers multiple languages, with English accounting for the vast majority. We do not apply additional preprocessing to the speech data, such as splitting into shorter segments; instead, we train directly on the original utterance lengths paired with their transcripts. We use 50 Hz, 128-bin mel-spectrograms extracted at a 24K Hz sampling rate, with a hop size of 480 samples (20 ms) and a window size of 1920 samples (80 ms). The STFT is computed with $n_{\text{fft}} = 1920$ using a Hann window, and mel filters span $[f_{\min}, f_{\max}] = [0, 12{,}000]$ Hz. Finally, we apply global mean–variance normalization to the mel features using precomputed statistics (mean $-4.92$, variance $8.14$). as both the input and reconstruction targets of our tokenizer, while first stacking every four consecutive frames to reduce the frame rate to 12.5 Hz for more efficient training. For waveform synthesis, we employ a Vocos-based (Siuzdak, 2024) vocoder to convert the mel spectrograms back to audio waveforms at 24K Hz.

**Training**  We train all models for a single epoch, corresponding to approximately 450K steps. For optimization, we adopt the AdamW (Loshchilov & Hutter, 2019) optimizer with $\beta_1 = 0.9$, $\beta_2 = 0.999$, a weight decay of 0.01, and a learning rate of $8 \times 10^{-5}$ with a warmup of 32K steps. To maximize GPU utilization and stabilize training over varying utterance lengths, we employ a *dynamic batch size* strategy: on each GPU, we pack utterances until the total duration reaches roughly 300 seconds of speech, corresponding to around 3750 tokens at our 12.5 Hz token rate. This ensures that each batch maintains a consistent computational footprint while preserving full-utterance training without segmentation.

## E  Limitations

While SiTok demonstrates strong performance on both speech reconstruction and downstream speech understanding tasks, it still falls short of continuous feature representations. Future work will focus on closing this performance gap. Furthermore, our diffusion-based decoder poses challenges for streaming generation. We are currently investigating fine-tuning strategies, such as chunk-wise AR diffusion, to enable low-latency or streaming outputs.

