# OpenReview forum: "Scaling Speech Tokenizers with Diffusion Autoencoders"
_ICLR.cc/2026/Conference — ICLR 2026 Poster_

### Official Review · Reviewer_z58S · 2025-10-27

**Soundness:** 3
**Presentation:** 3
**Contribution:** 3
**Rating:** 6
**Confidence:** 3

**Summary:**

The paper proposes several improvements to audio tokenizer training.
Flow matching objective replaces the adversarial training, CTC loss to enforce semantic preservation, decoder finetuning, token CFG.

**Strengths:**

Auxiliary CTC loss is a good idea and is shown to be very useful.
The flow matching simplifies and speeds up training, and introduces stochastisity to decoding, which might improve audio quality (rather than a deterministic decoder).
Decoder finetuning and Token CFG are intuitive ideas, that show benefits in practice.
Clear writing and multiple ablations.

**Weaknesses:**

Data is inhouse and not described in detail. Harder to reproduce or generalize to experiments done with different data mixtures.
Results are shown for just one dataset - SeedTTS.

**Questions:**

Table 5: why are the understanding metrics on Loss R / R+D identical? aren’t you training the encoder/decoder jointly?

Why did you choose to make the encoder causal, and the decoder bidirectional?

How do you project and predict audio? (e.g. project audio to the encoder, project the noise audio to the decoder, predict the audio)

How about adding the raw audio WER/Sim/UTMOS results?

---

> ### Author Response · Authors · 2025-11-24
> **Response to reviewer z58S**
>
> First of all, we want to thank the reviewer for your careful reading and providing a lot of constructive comments! Below we address the concerns mentioned in the review.
>
> ## Weakness
>
> ```Weakness 1: Data is inhouse and not described in detail. Harder to reproduce...```
>
> We thank the reviewer for raising this concern. We confirm that we will open-source both the full training code and the pretrained model checkpoints (trained on public, research-only datasets) after the paper decision to further support reproducibility. For the public model, we will provide detailed data description. In the revised version, we have also added detailed pseudo-code for the model implementation and the full training loop in Appendix D to make the method easier to follow and re-implement.
>
> ## Question
>
> ```Question 1: Table 5: why are the understanding metrics on Loss R / R+D identical? aren't you training the encoder/decoder jointly?```
>
> Thank you for catching this. In Table 5, the "R" and "R + D" rows differ only in the decoder: "R" is trained with a regression loss, while "R + D" further fine-tunes only the decoder with the diffusion objective, keeping the **encoder and VQ module frozen**. Our downstream understanding metrics (LLM-based ASR, ER, SV, KS) depend **only on the encoder and discrete tokens**, and do not use the decoder outputs. Therefore, by design, the understanding metrics for R and R + D are identical, while reconstruction metrics (WER/SIM/UTMOS) change. We will clarify this setting explicitly in the table caption and text to avoid confusion.
>
> ```Question 2: Why did you choose to make the encoder causal, and the decoder bidirectional?```
>
> Our design choice is mainly driven by the nature of the diffusion decoder and the different deployment constraints on encoding vs. decoding.
>
> Because the decoder is a diffusion model that reconstructs mel-spectrograms, we deliberately make it bidirectional to fully exploit future context and provide a non-streaming upper bound on the achievable quality at 0.2 kbps. In contrast, we choose a causal encoder so that SiTok can produce tokens in an online/streaming fashion, which is important for real-time speech interaction and speech LLM pretraining. In practical streaming scenarios, one can still use chunked or limited-lookahead decoding with the current decoder, while designing a fully causal diffusion decoder is an interesting but orthogonal direction that we leave for future work.
>
> ```Question 3: How do you project and predict audio? (e.g. project audio to the encoder, project the noise audio to the decoder, predict the audio)```
>
> We operate in the mel-spectrogram domain rather than directly on waveforms. Concretely, we first convert raw audio to a **50 Hz mel-spectrogram**, and then stack every 4 consecutive frames into one to obtain a **12.5 Hz mel representation**, which is fed into the Transformer encoder. The encoder outputs are quantized to discrete codes and mapped back to token embeddings, which condition the diffusion decoder. During training, we add Gaussian noise to the **12.5 Hz stacked mel** to obtain the noisy input for the diffusion decoder and predict the clean mel (or velocity) at that resolution. At inference time, we start from pure noise in the **12.5 Hz stacked mel**, run the diffusion sampler conditioned on the tokens to obtain a clean 12.5 Hz stacked mel, **unstack it back to 50 Hz**, and finally pass it through a vocoder to generate the waveform.
>
> ```Question 4: How about adding the raw audio WER/Sim/UTMOS results?```
>
> Of course! We have add the raw audio WER/SIM/UTMOS results to the revised manuscript in Table 1. We also provide the results here for your reference (we omit the results for the baselines for brevity):
>
> | Model | FPS/TPS | WER (↓) | SIM (↑) | UTMOS (↑) |
> | :--- | :---: | :---: | :---: | :---: |
> | Ground Truth | - | 2.14 | 0.730 | 3.53 |
> | **SiTok (CN=1)** | 12.5/12.5 | 4.06 | 0.641 | 3.44 |
> | &nbsp;&nbsp;*+ Decoder Finetuning* | 12.5/12.5 | 3.79 | 0.682 | 3.48 |
> | &nbsp;&nbsp;*+ Token CFG* | 12.5/12.5 | 3.34 | 0.635 | 3.60 |
> | **SiTok (CN=2)** | 12.5/25 | 3.17 | 0.658 | 3.44 |
> | **SiTok (CN=4)** | 12.5/50 | 2.80 | 0.660 | 3.46 |
>
> ---
>
> Thanks again for your constructive comments. We would be grateful if we could hear your feedback regarding our answers to the reviews. We would be happy to answer and discuss if you have further comments.

---

### Official Review · Reviewer_WeZq · 2025-10-31

**Soundness:** 3
**Presentation:** 3
**Contribution:** 2
**Rating:** 4
**Confidence:** 4

**Summary:**

This paper proposes SiTok, a diffusion-based speech tokenizer that achieves high compression ratios while maintaining high-fidelity speech reconstruction. A CTC loss–based semantic regularization method is introduced to enhance the semantic richness of quantized representations. In addition, the paper explores shortcut fine-tuning to accelerate diffusion inference. Extensive experiments demonstrate that SiTok delivers superior reconstruction and downstream understanding performance at low bitrates, and comprehensive ablation studies confirm the effectiveness of each module.

**Strengths:**

1.**Strong performance at extremely low bitrates.** The proposed SiTok achieves excellent results in both speech reconstruction quality and downstream understanding tasks at very low bitrates. In reconstruction evaluation, SiTok at 200 bps reaches a SIM score of 0.682, with its UTMOS and WER metrics are comparable to those of previous SOTA codec models. On downstream understanding tasks such as LLM based ASR, ER, SV, and KS, SiTok exhibits rich and effective semantic representations.

2. **Comprehensive and well-analyzed ablation studies.** The paper provides extensive ablation experiments that validate the effectiveness of each core component, including model scaling, accelerated diffusion strategies, CTC loss weighting, and reconstruction refinement. The corresponding analyses are clear and insightful.

3. **Exploration of acceleration strategies for diffusion-based tokenizers.** The exploration of shortcut fine-tuning and a light-weight diffusion head offers an solution to accelerate the diffusion decoding process while maintaining high reconstruction quality.

**Weaknesses:**

1.The overall design of the proposed speech tokenizer lacks novelty. (1) Prior works such as LaDiffCodec [1]
and TadiCodec [2] have already explored diffusion-based decoders to enhance reconstruction quality. (2)
Similarly, models like Baichuan Audio Tokenizer [3] and XY-Tokenizer [4] incorporate ASR-based
supervision to enrich semantic representations after quantization, although they adopt LLM-based
decoders rather than CTC loss.

2.While the paper emphasizes SiTokʼs strong reconstruction quality at high compression rates, it omits
important objective metrics such as STOI and PESQ, which weakens the persuasiveness of the evaluation.

3.Although several strategies for accelerating diffusion inference are explored, the paper lacks concrete
measurements and comparisons of reconstruction speed, such as RTF or related latency metrics.

[1] Yang, Haici, Inseon Jang, and Minje Kim. "Generative de-quantization for neural speech codec via latent
diffusion." ICASSP 2024-2024 IEEE International Conference on Acoustics, Speech and Signal Processing
(ICASSP). IEEE, 2024

[2] Wang, Yuancheng, et al. "Tadicodec: Text-aware diffusion speech tokenizer for speech language
modeling." arXiv preprint arXiv:2508.16790 (2025).

[3] Li, Tianpeng, et al. "Baichuan-audio: A unified framework for end-to-end speech interaction." arXiv
preprint arXiv:2502.17239 (2025).

[4] Gong, Yitian, et al. "XY-Tokenizer: Mitigating the Semantic-Acoustic Conflict in Low-Bitrate Speech
Codecs." arXiv preprint arXiv:2506.23325 (2025).

**Questions:**

1.While the paper focuses on reconstruction quality and semantic richness of the speech tokenizer, has the model been evaluated on TTS downstream tasks? Such results could better demonstrate its applicability to
generative scenarios.

2.The reconstructed audio achieves remarkably high SIM scores, even at 200 bps. Could the authors clarify
which component primarily contributes to this improvement? Or does the large-scale training data (2M
hours) play a dominant role in boosting SIM performance?

3.The encoder is causal, whereas the decoder is not. The authors briefly mention this limitation, but I am
curious whether converting the diffusion-based decoder into a causal one would lead to significant
degradation in reconstruction quality, as streaming applications are of growing interest to the community.

Here are some suggestions:

1.The training configuration is insufficiently detailed. It would be helpful to include information such as batch
size and the number of GPUs used.

2.The paper frequently mentions decoder fine-tuning, which seems to correspond to a two-stage training
scheme. The authors are encouraged to clarify and include these details in the training description.

---

> ### Author Response · Authors · 2025-11-24
> **Response to reviewer WeZq (Part 1/3)**
>
> First of all, we want to thank the reviewer for your careful reading and providing a lot of constructive comments! Below we address the concerns mentioned in the review.
>
> ## Weakness
>
> ```Weakness 1: The overall design of the proposed speech tokenizer lacks novelty...```
>
> We very appreciate the reviewer for pointing out the connections to LaDiffCodec, TaDiCodec, Baichuan Audio Tokenizer, and XY-Tokenizer. These are all highly relevant and inspiring works, and we will better highlight them in the revised related-work section.
>
> At the same time, our goal in SiTok is not to introduce diffusion or ASR supervision in isolation, but to propose and systematically study a unified tokenizer paradigm that jointly addresses (i) extreme compression, (ii) high-fidelity reconstruction, and (iii) semantic-rich representations for speech language modeling. Concretely, our contributions differ from prior work in several important aspects:
>
> **(1)** **Compared with LaDiffCodec** and other diffusion-based tokenizers (e.g., CosyVoice S3 tokenizer), which adopt a two-stage pipeline: (i) learn latents/quantizer, then (ii) train a separate diffusion decoder on fixed latents, SiTok is a end-to-end diffusion autoencoder, jointly optimizing encoder + VQ + diffusion decoder.
>
> We explicitly reproduced the two-stage design in Table 5 (R+D): it **degrades both reconstruction (SIM 0.641 to 0.587, WER 4.06 to 4.73)** and **downstream unserstanding tasks (ASR 4.95 to 6.06)** compared to our end-to-end approach. And for **TaDiCodec**, it requires **text as a condition** for the diffusion decoder (essentially a text-aware TTS-style decoder). SiTok is a **pure audio diffusion autoencoder** that does not rely on text and is suitable for unlabeled or multimodal LLM pretraining. TaDiCodec was also released on arXiv on Aug 22, after the ICLR submission deadline; per ICLR policy, comparison is not required, though we still discuss it in related work.
>
> **(2)** For Beichuan-Audio tokenizer and XY-Tokenizer, although they also incorporate ASR-based supervision to enrich semantic representations, they still rely on an additional ASR model (Whisper) as a semantic encoder to extract latent features. In contrast, **SiTok learns representations directly from raw speech**, while the **CTC loss serves purely as a lightweight regularizer** applied on the quantized tokens, without relying on any external ASR encoder.
>
> We appreciate the reviewer's suggestion. We will incorporate these discussions in the revised version of the paper to provide clearer comparisons and a more comprehensive analysis.
>
> ```Weakness 2: About objective metrics such as STOI and PESQ.```
>
> We thank the reviewer for the suggestion. In our work, we focus primarily on generation-oriented metrics rather than classical signal-level metrics such as STOI and PESQ, because the goal of SiTok is to support speech language modeling and downstream understanding tasks, where generation fidelity and semantic consistency are more relevant than signal distortion scores.
>
> In addition, to better reflect perceptual reconstruction quality, we provide a demo page [https://sitok-demo.github.io/](https://sitok-demo.github.io/) and subjective listening test results (in terms of CMOS) that better reflect reconstruction quality.
>
> | **System**                | **CMOS**          |
> |---------------------------|-------------------|
> | Ground Truth             | **+0.65 ± 0.12**  |
> | Mimi | -1.65 ± 0.22 |
> | WavTokenizer | -1.28 ± 0.36 |
> | BiCodec | -0.88 ± 0.15 |
> | **SiTok (CN=1)**         | **0.00**          |
>
> ```Weakness 3: Lacks concrete measurements and comparisons of reconstruction speed, such as RTF or related latency metrics.```
>
> Thank you for the suggestion. We agree that reporting wall-clock efficiency metrics, such as real-time factor (RTF), would further clarify the practical benefits. We provide the **RTF metrics for both SiTok reconstruction and the SiTok-based TTS system**:
>
> - For reconstruction, SiTok achieves an RTF of **0.041/0.024/0.013** with **16/8/4** steps for diffusion decoding.
>
> - We also build a SiTok-based TTS system and compare with SOTA AR-based TTS models includes CosyVoice 2, SparkTTS, and Llasa. The results are shown in the following table, our model achieves the best **RTF of 0.234** (using 16 steps for diffusion decoding) among all the models, demonstrating that our tokenizer is sufficiently efficient for practical speech language modeling applications:
>
> | Model | FPS/TPS | WER (↓) | SIM (↑) | RTF (↓) |
> |-------|---------|--------|--------|--------|
> | CosyVoice 2 | 25/25 | 2.89 | 0.66 | 0.455 |
> | SparkTTS | 12.5/12.5 | 2.50 | 0.57 | 0.601 |
> | Llasa | 50/50 | 3.94 | 0.58 | 0.422 |
> | SiTok-AR-TTS | 12.5/12.5 | 2.46 | 0.64 | 0.234 |
>
>
> *Note: RTF is measured on a single A100 GPU, averaging 10 runs of reconstruction or generating a 10s speech sample.*

---

> > ### Author Response · Authors · 2025-11-24
> > **Response to reviewer WeZq (Part 2/3)**
> >
> > ## Question
> >
> > ```Question 1: TTS reuslts```
> >
> > we appreciate the reviewer's suggestion and have conducted an additional TTS experiment to directly evaluate SiTok in a text-to-speech pipeline. Specifically, **we train a 0.5B LLM-based TTS model (initialized from Qwen2.5-0.5B) that autoregressively predicts SiTok tokens from text**, followed by our diffusion decoder to synthesize mel-spectrograms. We train this TTS model on 100k hours of the Emilia dataset to align with the baselines we compare against. More architectural and training details **for the TTS model will be provided in the revised appendix**.
> >
> > Evaluated on the SeedTTS test-en set, this system (denoted as SiTok-AR-TTS) achieves **WER: 2.46, SIM: 0.64, RTF: 0.234**.
> >
> > We also compare with SOTA AR-based TTS models includes CosyVoice 2, SparkTTS, and Llasa. The results are already shown in **weakness 3**.
> >
> > We observe that
> > - SiTok-AR-TTS delivers comparable or better WER and SIM than the baselines.
> > - We find that SiTok-AR-TTS achieves an even lower WER than SiTok reconstruction, a trend observed in other leading TTS systems. This is likely because the speech tokens are directly guided by the text.
> > - The RTF results further highlight a key advantage of our design: because **SiTok operates at 12.5 Hz**, the text-to-token autoregressive decoder processes a **much shorter sequence** than tokenizers running at 25–50 Hz, leading to substantially lower RTF.
> >
> > The additional TTS experiments demonstrate that SiTok can be directly applied to speech generation while maintaining strong performance in reconstruction and downstream understanding. We hope these results address your concerns.
> >
> > ```Question 2: The reconstructed audio achieves remarkably high SIM scores, even at 200 bps. Could the authors clarify which component primarily contributes to this improvement? Or does the large-scale training data (2M hours) play a dominant role in boosting SIM performance?```
> >
> > This is an excellent question, and we are happy to provide a deeper analysis of the key factors that contribute to the remarkably high SIM scores.
> >
> > **(1) Modeling perspective.** Both CTC semantic regularization and the diffusion decoder play crucial roles. As shown in Table 3 and Table 5, removing CTC regularization on the quantized latents increases WER from 4.06 to 33.0 and reduces SIM from 0.641 to 0.495 at 12.5 Hz / 0.2 kbps, even though the data, architecture, and training setup remain identical. This demonstrates that enforcing a CTC objective on the discrete codes not only improves intelligibility but also helps allocate the limited token capacity toward linguistically and speaker-relevant features—an essential property in the extreme low-bitrate regime. Table 5 further shows that replacing the diffusion objective with a simple regression loss (L1 on mel) leads to lower SIM (0.587 vs. 0.641) and higher ASR errors. Moreover, fine-tuning only the decoder with diffusion on top of a regression-pretrained model cannot close this gap, indicating that joint diffusion training is necessary.
> >
> > **(2) Data perspective.** In fact, in our early preliminary experiments using a much smaller dataset (~100k hours), we found that reconstruction quality was already close to the 2M-hour setting, whereas downstream understanding tasks (e.g., ASR) showed a much larger performance gap. This suggests that learning speaker timbre and acoustic details is comparatively easier, while **achieving strong semantic representation benefits much more from data scaling**.

---

> > > ### Author Response · Authors · 2025-11-24
> > > **Response to reviewer WeZq (Part 3/3)**
> > >
> > > ```Question 3: Decoder is non-casual and how to support streaming inference.```
> > >
> > > We fully agree with the reviewer that streaming inference is an important and increasingly relevant capability. In our current design, we use a causal encoder and a non-causal diffusion decoder, as diffusion models generally benefit from limited future context to improve reconstruction fidelity. Imposing strict causality on the decoder would likely degrade quality, particularly at extremely low bitrates, where future context helps disambiguate linguistic content and speaker characteristics.
> > >
> > > That said, supporting fully streaming decoding is a valuable direction. We plan to explore AR-Diffusion or chunk-wise autoregressive diffusion as potential solutions, and we will clarify this future research path in the revised discussion section.
> > >
> > > Importantly, even with a non-causal decoder, our causal encoder and very low token rate (12.5 Hz) already enable a practical streaming-style workflow: speech can be encoded causally into tokens, and decoding can be performed on short overlapping token chunks with minimal look-ahead. Diffusion then operates on these short mel segments, keeping the algorithmic latency low.
> > >
> > > In addition, our exploration of shortcut modeling significantly reduces diffusion latency, achieving high-quality reconstruction with only 2–4 sampling steps, which further improves the practicality of near-real-time decoding. We also report the RTF results for both reconstruction and downstream TTS in **weakness 3**.
> > >
> > > ```Suggestion 1: The training configuration is insufficiently detailed. It would be helpful to include information such as batch size and the number of GPUs used.```
> > >
> > > Thank you for the suggestion. We add the missing training configuration details, including batch size, number of GPUs, and other key hyperparameters, in the revised manuscript.
> > >
> > > In addition, we include more details:
> > > - Detailed architectural specifications and pseudo-code for the encoder, codebook, diffusion decoder, and CTC regularizer (Appendix D.1, page 19 to 21),
> > > - Pseudo-code for the core training loop of the end-to-end diffusion autoencoder (Appendix D.2, page 22), and
> > > - More details about the training and the data-processing pipeline (Appendix D.3, page 22 to 23).
> > >
> > > These additions should provide sufficient clarity and reproducibility for future researchers.
> > > We also confirm that we will open-source both the code and the pretrained model checkpoints (on public, research-only datasets) upon the decision of the paper.
> > >
> > >
> > > ```Suggestion 2: Details about decoder fine-tuning.```
> > >
> > > We appreciate the opportunity to clarify this point. Our training procedure consists of:
> > >
> > > - **A main end-to-end training stage**, where the encoder, VQ module, diffusion decoder, and CTC decoder are trained jointly with the combined reconstruction, CTC, and VQ losses (Section 2.2 and 3.1).
> > >
> > > - **An optional refinement stage**, where we **freeze the encoder and VQ** and only further train the diffusion decoder on the fixed discrete tokens. This stage includes (i) the "decoder fine-tuning"  used to specialize the decoder for high-fidelity mel reconstruction, and (ii) the "shortcut fine-tuning" objective used to adapt the decoder to very low-step sampling.
> > >
> > > Thus, "decoder fine-tuning" is not a separate tokenizer in itself but an **additional optional refinement** on top of the end-to-end diffusion autoencoder.
> > >
> > > ---
> > >
> > > Thanks again for your constructive comments. We would be grateful if we could hear your feedback regarding our answers to the reviews. We would be happy to answer and discuss if you have further comments.

---

### Official Review · Reviewer_5u8g · 2025-10-31

**Soundness:** 4
**Presentation:** 2
**Contribution:** 3
**Rating:** 6
**Confidence:** 4

**Summary:**

This paper focuses on an important topic in the audio domain—the design of speech tokenizers. By scaling both the model size and the amount of training data, the authors build a highly performant tokenizer and provide a detailed discussion of its training strategies from multiple perspectives.

**Strengths:**

- The paper scales the speech tokenizer up to a 1.6B-parameter model trained on 2 million hours of speech data, effectively enhancing model performance and achieving state-of-the-art results.

- The authors conduct comprehensive ablation studies on various aspects of the tokenizer’s training process and offer insightful observations on effective training techniques.

**Weaknesses:**

- The current version appears to lack qualitative analyses or examples of actual reconstructed speech. Since audio reconstruction fidelity is critical for evaluating a speech tokenizer, it would be highly valuable to include qualitative results (e.g., audio samples) to better demonstrate the model’s effectiveness. **If such results are not presented in the rebuttal, I would consider decreasing my score.**

- The main improvements in this work seem to stem from scaling the model and training data, while many methodological components follow prior work. Nevertheless, the systematic analysis and strong pretraining effort provide meaningful progress for the field.

**Questions:**

It is unclear whether the authors plan to release the model checkpoints or pretrained weights. Without open-sourcing, the paper’s overall impact and reproducibility would be significantly limited. Clarification on this point would be appreciated.

---

> ### Author Response · Authors · 2025-11-24
> **Response to reviewer 5u8g**
>
> First of all, we want to thank the reviewer for your careful reading and providing a lot of constructive comments! Below we address the concerns mentioned in the review.
>
> ## Weakness
>
> ```Weakness 1: The current version appears to lack qualitative analyses or examples of actual reconstructed speech...```
>
> Thank you for highlighting the importance of qualitative evidence for reconstructed speech. We fully agree that subjective inspection is essential for evaluating a speech tokenizer. In response, we have added a comprehensive **demo page with qualitative comparisons** (in the link: [https://sitok-demo.github.io/](https://sitok-demo.github.io/)), including:
> - (1) Reconstruction results for SiTok and baselines, we choose some in-the-wild and expressive speech samples to better showcase the reconstruction quality.
> - (2) **Additional TTS results** with SiTok tokens (as discussed in Weakness 1 of Reviewer XL4L).
>
> In addition, we have conducted **subjective listening tests**, reporting CMOS (comparative Mean Opinion Score) and results on reconstructed speech. **The results and more details about the listening test are added to the appendix of the revised paper (Appendix C.4)**.
>
> | **System**                | **CMOS**          |
> |---------------------------|-------------------|
> | Ground Truth             | **+0.65 ± 0.12**  |
> | Mimi | -1.65 ± 0.22 |
> | WavTokenizer | -1.28 ± 0.36 |
> | BiCodec | -0.88 ± 0.15 |
> | **SiTok (CN=1)**         | **0.00**          |
>
> We hope these reconstructed speech samples and subjective assessments address the reviewer's concern.
>
> ```Weakness 2: The main improvements in this work seem to stem from scaling the model and training data, while many methodological components follow prior work. Nevertheless, the systematic analysis and strong pretraining effort provide meaningful progress for the field.```
>
> We thank the reviewer for acknowledging the value of our systematic analysis and large-scale pretraining effort! We want tot briefly restate our core contribution and clarify our distinctions from prior work.
>
> In this work, our main motivation is to provide a **scalable paradigm** for training a tokenizer that simultaneously achieves high-quality reconstruction and strong downstream speech language modeling performance, at an **extremely low token rate (12.5 Hz)** and **200 bps bitrate** for efficient speech language modeling.
>
> We adopt an **end-to-end diffusion autoencoder** with **semantic regularization via a CTC decoder**, which jointly learns semantic-rich representations through supervised learning while enabling high-fidelity audio reconstruction through diffusion. This differs from prior two-stage diffusion tokenizers, which first train the encoder and quantizer and then train a diffusion decoder on fixed latents. Our experiments show that such end-to-end joint optimization is essential, compared with the two-stage alternative, for achieving both strong reconstruction quality and robust downstream performance. In addition, our extensive ablation studies provide further insights into the key design choices that contribute to SiTok’s overall effectiveness.
>
> ## Question
>
> ```Question 1: It is unclear whether the authors plan to release the model checkpoints or pretrained weights...```
>
> We appreciate the reviewer's emphasis on reproducibility. **We confirm that we will open-source both the inference code and the pretrained model checkpoints (on public, research-only datasets)** upon the decision of the paper.
>
> To further support reproducibility, we have added:
>
> - **Detailed architectural specifications and Pseudo-code** for the encoder, codebook, diffusion decoder, and CTC regularizer (Appendix D.1, page 19 to 21),
> - **Pseudo-code outlining the core training loop** for the end-to-end diffusion autoencoder (Appendix D.2, page 22), and
> - **Training hyperparameters and data-processing steps** (Appendix D.3, page 22 to 23)
>
> in the the revised manuscript, feel free to check it. These additions should make it significantly easier for others to reproduce our results or extend the framework.
>
> ---
>
> Thanks again for your constructive comments. We would be grateful if we could hear your feedback regarding our answers to the reviews. We would be happy to answer and discuss if you have further comments.
>
> ---
>
> References:
>
> [1] Défossez, Alexandre, et al. "Moshi: a speech-text foundation model for real-time dialogue." arXiv preprint arXiv:2410.00037 (2024).
>
> [2] Ji, Shengpeng, et al. "Wavtokenizer: an efficient acoustic discrete codec tokenizer for audio language modeling." arXiv preprint arXiv:2408.16532 (2024).
>
> [3] Ye, Zhen, et al. "Codec does matter: Exploring the semantic shortcoming of codec for audio language model." Proceedings of the AAAI Conference on Artificial Intelligence. Vol. 39. No. 24. 2025.

---

### Official Review · Reviewer_mPsW · 2025-11-01

**Soundness:** 2
**Presentation:** 3
**Contribution:** 2
**Rating:** 4
**Confidence:** 3

**Summary:**

The paper introduces the Speech Diffusion Tokenizer (SiTok), a diffusion-based autoencoder that jointly optimizes vector quantization and waveform reconstruction. To align its discrete codes with linguistic information, SiTok incorporates semantic regularization via a CTC decoder. To enhance efficiency, the authors apply shortcut fine-tuning and other acceleration techniques to reduce diffusion steps without compromising quality. Trained on 2 million hours of speech with 1.6B parameters, the paper claims that SiTok is effective even at an extremely low token rate (12.5 Hz, 0.2 kbps) and outperforms previous baselines.

**Strengths:**

- The paper’s strength lies in its simple framework consists of a diffusion autoencoder, vector quantization, and CTC supervision. While not conceptually novel, it’s highly practical. The auxiliary CTC loss proves crucial (WER drops from 33.0 to 4.06, Table 3). Refinements like shortcut fine-tuning and classifier-free guidance further boost efficiency, reflecting thoughtful system design.

- The experiments are comprehensive. Table 5 explores eight design factors (e.g., loss type, codebook settings, frame rate) and evaluates both reconstruction (WER, SIM, UTMOS) and understanding (ER, SV, KS, ASR) tasks.

- At 0.2 kbps (12.5 Hz), SiTok delivers competitive performance in a highly compressed regime. With a single codebook, it achieves WER 4.06 and SIM 0.641

**Weaknesses:**

- The paper prefers mel-spectrograms over raw waveforms without strong justification. Modern architectures can handle raw waveforms efficiently, and prior speech tokenizers (e.g., EnCodec, WavTokenizer, BigCodec, Mimi) operate directly on them successfully. Claims that waveforms are inefficient or adversarially complex are unconvincing, especially given that adversarial methods have been applied effectively in prior work. Mel-spectrograms are lossy and require a separate vocoder (Vocos), yet no comparison to waveform-based models is provided.

- The method relies on a single VQ codebook, despite evidence (Table 5) that residual VQ (RVQ) substantially improves performance (WER 4.06 → 2.50 with 8 codebooks). Leading tokenizers employ RVQ to avoid bottlenecks at low token rates. The paper provides no rationale for using a single codebook, ignores alternatives such as FSQ.

- Diffusion-based speech tokenization is not novel; prior work includes CosyVoice, Vevo, FireRedTTS, NaturalSpeech 3, DiTAR, and OZSpeech. The claimed novelty of end-to-end training with CTC supervision is not clearly superior to established two-stage approaches.

- The regression loss achieves a WER of 4.66 compared to 4.06 for diffusion, so the improvement is far from marginal. This raises questions about the practical benefit of using diffusion loss over a simpler L1 loss.

- The paper is titled "SCALING Speech Tokenizers" but Table 4 shows scaling doesn't work: the 1.61B (XL) model performs worse than the 1.12B (L) model on key metrics (ASR: 5.07 vs 4.95; SV: 14.7 vs 13.8).

**Questions:**

- What is the real-time factor (RTF) and actual inference latency in milliseconds for the method compared to single-pass baselines?

- How does task performance change if the CTC decoder is scaled proportionally (e.g., 24 layers for the XL model) rather than keeping it fixed at 4 layers?

- Was Finite Scalar Quantization (FSQ) evaluated as an alternative to vector quantization (VQ)?

- Can statistical significance tests be provided for the reported performance gap, given that the improvement is minimal?

---

> ### Author Response · Authors · 2025-11-24
> **Response to reviewer mPsW (Part 1/3)**
>
> First of all, we want to thank the reviewer for your careful reading and providing a lot of constructive comments! Below we address the concerns mentioned in the review.
>
> ## Weakness
>
> ```Weakness 1: The paper prefers mel-spectrograms over raw waveforms without strong justification...```
>
> We acknowledge that our current discussion of mel-spectrograms versus raw waveforms can be more precise. Our goal is not to argue that waveform-based tokenizers are inherently inferior, recent systems such as EnCodec, BigCodec, and Mimi clearly demonstrate that raw-waveform modeling is effective. In our setting, we adopt mel-spectrograms primarily as a **practical and scalable design choice**.
>
> - At 24 kHz, waveform sequences are several hundred times longer than our 50 Hz mel representation (which is further stacked to 12.5 Hz for transformer-based encoder/decoder efficiency), whereas waveform models rely on **substantial strided/downsampling front-ends to keep computation manageable**. Training a 1.6B-parameter diffusion decoder on 2M hours of audio **without aggressive chunking** is therefore much more tractable in the mel domain. Mel-spectrograms allow us to train on **full-length utterances** and to maintain an extremely low token rate (12.5 Hz) in a tractable way.
> - We also find that large-scale adversarial training on raw waveforms typically requires multiple discriminators and carefully tuned multi-scale objectives to reach high acoustic quality, while a mel+vocoder pipeline **decouples fine-grained waveform detail into the vocoder and allows the tokenizer to focus on learning a compact, semantically useful representation**. In our preliminary experiments, diffusion on mel-spectrograms provided more stable optimization under the same compute budget.
>
> - *Our core contributions, a diffusion autoencoder jointly optimizing quantization and reconstruction, semantic regularization via a CTC decoder, and a systematic scaling/ablation study at 12.5 Hz and 0.2 kbps – are orthogonal to the choice of acoustic target*. The same ideas can be applied to waveform-based tokenizers, which we see as a promising direction for future work.
>
> ```Weakness 2: The paper provides no rationale for using a single codebook, ignores alternatives such as FSQ.```
>
> (1) We appreciate this comment and agree that our motivation for focusing on a single codebook in the main results should be better explained. *Our primary goal in this paper is to study scaling under extreme compression: 0.2 kbps at 12.5 Hz with CN=1, which is substantially more aggressive than most existing tokenizers*. We therefore chose the single-codebook configuration as the default operating point to demonstrate that SiTok can already deliver strong reconstruction (WER 4.06, SIM 0.641) and understanding performance (LLM ASR 4.95) under this highly constrained bitrate. That said, we do not claim that a single codebook is optimal. In fact, Table 5 explicitly investigates increasing the number of codebooks via RVQ (CN = 1 to 8), and we observe substantial gains: the reconstruction WER improves from 4.30 to 2.50, and both ASR and SV metrics are consistently boosted as bitrate increases.  This confirms that our framework naturally supports the standard RVQ strategy, and that SiTok can trade bitrate for fidelity in a controlled manner.
>
>
> (2) Comparison with FSQ: While our core contribution, the generative scaling of speech tokenization, is orthogonal to the specific quantization method, our framework can naturally support FSQ or other VQ alternatives (BSQ, LFQ...). We would like to conduct an additional experiment with FSQ to address the reviewer's suggestion. As shown in the table below, standard VQ consistently outperforms FSQ in our large-scale setting. We attribute this to two factors:
> - **Expressiveness**: The learnable embeddings in VQ capture richer acoustic and semantic information compared to the fixed scalar projections of FSQ.
> - **Stability at Scale**: FSQ is often employed to improve training stability. However, under our training seeting (large batch sizes, and EMA updates), we observe that our standard VQ is already highly stable with codebook usage exceeding 95%.
> Consequently, since we do not suffer from codebook collapse, the additional expressiveness of VQ yields better performance than FSQ. We include this comparison and details about the experiment in the revised of the paper (**Appendix C.4, page 19 to 20**) to demonstrate that while SiTok is compatible with FSQ, VQ is the empirically superior choice for our setup.
>
> | Model | WER (↓) | SIM (↑) | UTMOS (↑) | CTC ASR (↓) | ASR (↓) | ER (↑) | SV (↓) | KS (↑) |
> | :--- | :---: | :---: | :---: | :---: | :---: | :---: | :---: | :---: |
> | **SiTok (with VQ, 32 dim)** | 4.06 | 0.641 | 3.44 | 9.50 | 4.95 | 63.5 | 13.8 | 96.9 |
> | **SiTok (with FSQ)** | 5.23 | 0.629 | 3.44 | 10.02 | 5.33 | 62.0 | 14.1 | 96.9 |

---

> > ### Author Response · Authors · 2025-11-24
> > **Response to reviewer mPsW (Part 2/3)**
> >
> > ```Weakness 3: Diffusion-based speech tokenization is not novel; prior work includes CosyVoice, Vevo, FireRedTTS, NaturalSpeech 3, DiTAR, and OZSpeech...```
> >
> > We would like to clarify the categorization of prior work. Among the systems mentioned by the reviewer, NaturalSpeech 3, DiTAR, and OZSpeech do not involve diffusion-based speech tokenizers. Other diffusion-based tokenizers proposed by CosyVoice, Vevo, and FireRedTTS typically adopt a **two-stage pipeline**, where the tokenizer is trained separately from the diffusion decoder. This separation *decouples representation learning from reconstruction*, and *the quantizer itself is not optimized jointly with the generative objective*.
> >
> > In contrast, **our work proposes an end-to-end diffusion autoencoder** in which the encoder, VQ codebook, and diffusion decoder are **jointly optimized**. This design ensures that discrete codes are simultaneously (i) highly compressed, (ii) reconstructable with high fidelity, and (iii) semantically aligned through the CTC regularizer. The goal is not to claim conceptual novelty of diffusion itself, but to show that such an integrated formulation scales effectively to large data size and operates at extremely low token rates, and produces tokens that support both reconstruction and downstream speech-language modeling.
> >
> > ```Weakness 4: This raises questions about the practical benefit of using diffusion loss over a simpler L1 loss.```
> >
> > Regarding the diffusion objective, our experimental evidence indicates that diffusion is not a marginal improvement over a regression loss under this extreme compression regime. As shown in Table 5, compared to an L1 regression objective (R), the diffusion objective (D) yields consistent gains across metrics: **WER: 4.66 → 4.06 (13% relative), SIM: 0.587 → 0.641 (9% relative), UTMOS: 3.28 → 3.44 (5% relative), LLM-based ASR WER: 6.06 → 4.95 (18% relative)**. Such relative improvements, especially in WER are generally regarded as substantial/considerable in the ASR community. Simply fine-tuning a regression-pretrained model with a diffusion decoder (R+D) does not recover this gap (WER 5.73), suggesting that **end-to-end diffusion training produces qualitatively different and stronger representations**.
> >
> > Subjectively, models trained only with regression losses exhibit clear *artifacts* and *over-smoothing*, resulting in perceptually *"muddy"* acoustics. While these degradations do not drastically affect WER, they noticeably reduce overall audio quality (SIM and UTMOS).
> >
> > ```Weakness 5 : The paper is titled "SCALING Speech Tokenizers" but Table 4 shows scaling doesn't work...```
> >
> > We appreciate the reviewer for drawing attention to the scaling results in Table 4. Our goal in this section is not to claim that "bigger is always better", but to characterize how reconstruction and understanding trade off as we scale the tokenizer under an extreme compression regime.
> >
> > Indeed, Table 4 reveals a nuanced picture: as we scale from S → B → L → XL, reconstruction metrics (WER, SIM, UTMOS) improve monotonically, with the XL model achieving the best reconstruction quality. However, downstream understanding tasks peak at the L configuration (1.12B), with the XL model exhibiting slightly worse ASR and SV despite its stronger acoustic fidelity.
> >
> > This suggests that, at 12.5 Hz and 0.2 kbps, excessive capacity may over-emphasize fine-grained acoustic detail at the expense of abstract, discriminative features that are most useful for speech understanding.
> >
> > We see this not as a failure of scaling, but as a key scaling insight: there is an optimal model size that best balances fidelity and semantic abstraction in the tokenizer, and pushing beyond that point can saturate or slightly degrade understanding performance. We will make this interpretation more explicit in the paper, emphasizing that our contribution is a systematic scaling analysis of diffusion-based speech tokenizers, rather than a claim that performance is strictly monotonic with parameter count. We will also clarify in the conclusion that we recommend the L configuration as the most effective compromise in this regime.

---

> > > ### Author Response · Authors · 2025-11-24
> > > **Response to reviewer mPsW (Part 3/3)**
> > >
> > > ## Question
> > >
> > > ```Question 1: About RTF.```
> > >
> > > Thank you for pointing out this missing piece of evaluation. **We provide the RTF for both SiTok reconstruction and downstream TTS.**
> > >
> > > - For reconstruction, SiTok achieves an RTF of **0.041/0.024/0.013** with **16/8/4** steps for diffusion decoding.
> > >
> > > - We also build a SiTok-based TTS system and compare with SOTA AR-based TTS models includes CosyVoice 2, SparkTTS, and Llasa. The results are shown in the following table, our model achieves the best **RTF of 0.234** (using 16 steps for diffusion decoding) among all the models, demonstrating that our tokenizer is sufficiently efficient for practical speech language modeling applications:
> > >
> > > | Model | FPS/TPS | WER (↓) | SIM (↑) | RTF (↓) |
> > > |-------|---------|--------|--------|--------|
> > > | CosyVoice 2 | 25/25 | 2.89 | 0.66 | 0.455 |
> > > | SparkTTS | 12.5/12.5 | 2.50 | 0.57 | 0.601 |
> > > | Llasa | 50/50 | 3.94 | 0.58 | 0.422 |
> > > | SiTok-AR-TTS | 12.5/12.5 | 2.46 | 0.64 | 0.234 |
> > >
> > > *Note: RTF is measured on a single A100 GPU, averaging 10 runs of reconstruction or generating a 10s speech sample.*
> > >
> > > ```Question 2: How does task performance change if the CTC decoder is scaled proportionally.```
> > >
> > > We appreciate this insightful suggestion. We intentionally keep the CTC decoder light-weight (4 layers) and fixed across model sizes for two reasons:
> > >
> > > - to ensure that CTC ASR performance primarily reflects the quality of the tokens rather than the capacity of the CTC head itself; and
> > > - to keep the additional inference cost for CTC decoding modest, since our main downstream evaluation relies on a separate LLM-based ASR model that already has substantial capacity.
> > >
> > > In other words, we view the CTC decoder mainly as a semantic regularizer during training and a diagnostic head at evaluation time, rather than as a fully scaled ASR model. Increasing its depth to, say, more layers could improve the raw CTC ASR numbers, but would also blur the distinction between "token quality" and "decoder capacity", and make it harder to interpret the scaling behavior of the tokenizer itself.
> > >
> > > That said, we agree that reporting how performance changes with a moderately larger CTC decoder would be informative. We provide an ablation where we increase the CTC decoder depth (from 4 to 8) to provide a quantitative analysis.
> > >
> > > | Model | WER (↓) | SIM (↑) | UTMOS (↑) | CTC ASR (↓) | ASR (↓) | ER (↑) | SV (↓) | KS (↑) |
> > > | :--- | :---: | :---: | :---: | :---: | :---: | :---: | :---: | :---: |
> > > | **SiTok (CTC decoder: 4 layers)** | 4.06 | 0.641 | 3.44 | 9.50 | 4.95 | 63.5 | 13.8 | 96.9 |
> > > | **SiTok (CTC decoder: 8 layers)** | 4.22 | 0.650 | 3.46 | 9.38 | 5.09 | 63.2 | 14.5 | 96.0 |
> > >
> > > The results show that increasing the CTC decoder depth from 4 to 8 layers brings only minor changes. A deeper decoder slightly improves CTC-ASR and SIM/UTMOS, but does not translate into consistent gains across downstream tasks—WER and ASR even show some degradation.
> > >
> > > ```Question 3: Compared with FSQ.```
> > >
> > > We answer the question in Weakness 2.
> > >
> > > ```Question 4: Can statistical significance tests be provided for the reported performance gap.```
> > >
> > > We appreciate this comment. Regarding objective metrics (e.g., WER, SIM), our evaluations are conducted on standard large-scale benchmarks. Given the deterministic nature of the decoding process and the extensive size of the test sets, run-to-run variance is negligible. Thus, the reported gaps, while sometimes small, are consistent and robust.
> > >
> > > However, we agree that statistical significance is crucial for subjective evaluation. To address this, we have provided additional subjective results complete with 95% Confidence Intervals (CI). We report the results in terms of CMOS (comparative Mean Opinion Score). These intervals confirm that the improvements in audio quality are statistically significant and clearly perceptible.
> > >
> > > | **System**                | **CMOS**          |
> > > |---------------------------|-------------------|
> > > | Ground Truth             | **+0.65 ± 0.12**  |
> > > | Mimi | -1.65 ± 0.22 |
> > > | WavTokenizer | -1.28 ± 0.36 |
> > > | BiCodec | -0.88 ± 0.15 |
> > > | **SiTok (CN=1)**         | **0.00**          |
> > >
> > >
> > > Feel free to check all details about the subjective evaluation in the **Appendix C.3 (from page 18 to 19)** of the revised manuscript.
> > >
> > > ---
> > >
> > > Thanks again for your constructive comments! We would be grateful if we could hear your feedback regarding our answers to the reviews. We would be happy to answer and discuss if you have further comments.

---

### Official Review · Reviewer_xL4L · 2025-11-03

**Soundness:** 2
**Presentation:** 3
**Contribution:** 3
**Rating:** 4
**Confidence:** 3

**Summary:**

This paper proposes Speech Diffusion Tokenizer (SiTok), a diffusion autoencoder-based speech tokenizer. It aims to achieve extreme compression, high-quality reconstruction, and effective semantic representations for speech language modeling simultaneously. SiTok addresses the limitations of existing speech tokenizers in balancing compression rate, reconstruction quality, and semantic representation by jointly learning semantic-rich representations and high-fidelity audio reconstruction.

**Strengths:**

1. Performance: SiTok achieves superior performance in both speech reconstruction and understanding tasks at extremely low bit rates (0.2kbps) and token rates (12.5Hz), outperforming several strong baselines. This highlights its potential for efficient speech representation and modeling.
2. Innovative Design: The combination of diffusion models and semantic regularization is a novel approach in speech tokenization. These innovations enable SiTok to learn representations that are both acoustically faithful and semantically meaningful.
3. Flexibility: The model's performance can be further enhanced by adjusting parameters such as model size and codebook configurations, offering flexibility for different applications.

**Weaknesses:**

1. Lack of Direct TTS Task Evaluation: While the paper demonstrates strong performance on various speech understanding tasks such as ASR, ER, KS, and SV, it does not directly evaluate the proposed SiTok model on TTS tasks. This is a significant gap, as TTS tasks have unique requirements and challenges that may not be fully addressed by the current evaluation metrics.
2. Lack of Detail on Multi-Codebook CTC Decoder Implementation: The paper does not provide specific details on which layer the CTC decoders are applied to in the multi-codebook setup. This lack of clarity can make it difficult for other researchers to reproduce and build upon the work.
3. Real-time Challenges: Despite efforts to accelerate decoding, diffusion models' iterative sampling steps may still pose latency issues for real-time applications. Further optimization is needed to improve real-time performance.
4. Resource Intensity: The large model size (1.6B parameters) requires substantial computational and storage resources for training and inference, potentially restricting its deployment on resource-constrained devices.

**Questions:**

1. Why is there no demo page?
2. Ablation study on CTC loss supervision:
  The ablation study shows significant improvements in both WER and similarity metrics when CTC loss supervision is applied. While the improvement in WER is understandable as it directly relates to the semantic alignment of the speech tokens with the text, the substantial increase in similarity is puzzling. Typically, semantic and acoustic aspects involve a trade-off, and enhancing one might come at the cost of the other. Could the authors provide more detailed experimental insights into why the similarity metric shows such a marked improvement? This would help in better understanding the model's behavior and the interplay between semantic and acoustic features.
3. Comparison with Tokenizers Trained for Understanding Tasks:
  Many of the tokenizers compared in the understanding tasks seem to be trained primarily for reconstruction tasks. The performance metrics on understanding tasks may not be as convincing as they could be. Could the authors provide additional comparative experiments with tokenizers that are specifically trained for understanding tasks, such as S3Tokenizer? This would help in assessing the true effectiveness of SiTok in the context of speech understanding.

---

> ### Author Response · Authors · 2025-11-24
> **Response to reviewer xL4L (Part 1/3)**
>
> First of all, we want to thank the reviewer for your careful reading and providing a lot of constructive comments! Below we address the concerns mentioned in the review.
>
> ## Weakness
>
> ```Weakness 1: Lack of Direct TTS Task Evaluation.```
>
> We sincerely thank the reviewer for pointing this out. First, we thank the reviewer for acknowledging the strong results of SiTok in reconstruction and downstream understanding. And we fully agree that demonstrating the usability of SiTok in a downstream TTS setup would further strengthen the paper. Our work focuses specifically on *speech tokenization*, and therefore, following standard practice in prior tokenizer works (e.g., BigCodec, WavTokenizer, SemantiCodec, StableCodec).  We evaluate tokenizers mainly via *reconstruction quality and semantic effectiveness*. These metrics have been broadly adopted as the primary indicators of a tokenizer's suitability for generation tasks.
>
> That said, we appreciate the reviewer's suggestion and have conducted an additional TTS experiment to directly evaluate SiTok in a text-to-speech pipeline. Specifically, **we train a 0.5B LLM-based TTS model (initialized from Qwen2.5-0.5B) that autoregressively predicts SiTok tokens from text**, followed by our diffusion decoder to synthesize mel-spectrograms. We train this TTS model on 100k hours of the Emilia dataset to align with the baselines we compare against. More architectural and training details **for the TTS model will be provided in the revised appendix**.
>
> Evaluated on the SeedTTS test-en set, this system (denoted as SiTok-AR-TTS) achieves **WER: 2.46, SIM: 0.64, RTF: 0.234**.
>
> We also compare with SOTA AR-based TTS models includes CosyVoice 2, SparkTTS, and Llasa. The results are shown in the following table:
>
> | Model | FPS/TPS | WER (↓) | SIM (↑) | RTF (↓) |
> |-------|---------|--------|--------|--------|
> | CosyVoice 2 [1] | 25/25 | 2.89 | 0.66 | 0.455 |
> | SparkTTS [2] | 12.5/12.5 | 2.50 | 0.57 | 0.601 |
> | Llasa [3] | 50/50 | 3.94 | 0.58 | 0.422 |
> | SiTok-AR-TTS | 12.5/12.5 | 2.46 | 0.64 | 0.234 |
>
> We observe that
> - SiTok-AR-TTS delivers comparable or better WER and SIM than the baselines.
> - We find that SiTok-AR-TTS achieves an even lower WER than SiTok reconstruction, a trend observed in other leading TTS systems. This is likely because the speech tokens are directly guided by the text.
> - The RTF results further highlight a key advantage of our design: because **SiTok operates at 12.5 Hz**, the text-to-token autoregressive decoder processes a **much shorter sequence** than tokenizers running at 25–50 Hz, leading to substantially lower RTF.
>
> The additional TTS experiments demonstrate that SiTok can be directly applied to speech generation while maintaining strong performance in reconstruction and downstream understanding. We hope these results address your concerns.
>
> ```Weakness 2: Lack of Detail on Multi-Codebook CTC Decoder Implementation.```
>
> We thank the reviewer for this question. To clarify, in our multi-codebook setting, the input to both the auxiliary CTC decoder and the main diffusion decoder is **the sum of the embeddings from all RVQ layers**.
>
> ```Weakness 3: Real-time Challenges.```
>
> We acknowledge the reviewer's concern regarding the inherent latency of diffusion models. Addressing this challenge was a key focus of our work.
>
> As detailed in **Section 2.3 and evaluated in Section 3.3.5 ("Efficient Decoding")**, we implemented and analyzed strategies specifically designed to mitigate this issue. Our experiments with Shortcut Fine-tuning (as shown in Figure 2) demonstrate that we can drastically reduce the number of inference steps to as few as 2 or 4, while still maintaining high reconstruction quality (low WER and high similarity).
>
> To quantify this efficiency, **we measured the Real-Time Factor (RTF) of our model**.
> - The tokenizer itself achieves an RTF of **0.041/0.024/0.013** with **16/8/4** steps for diffusion decoding.
> - More importantly, when integrated into a downstream TTS task (as disscussed in Weakness 1), the entire system maintains an RTF of **0.234** (with 16 steps for diffusion decoding), demonstrating that our tokenizer is sufficiently efficient for practical speech language modeling applications.
>
> *Note: RTF is measured on a single A100 GPU, averaging 10 runs of reconstruction  or generating a 10s speech sample.*

---

> > ### Author Response · Authors · 2025-11-24
> > **Response to reviewer xL4L (Part 2/3)**
> >
> > ```Weakness 4: Resource Intensity.```
> >
> > We appreciate the reviewer's concern regarding the resource requirements of our largest 1.61B parameter model.
> >
> > Our work aims to develop a foundational tokenizer suitable for **large Speech Language Models (SLMs)**. In this context, larger tokenizer capacities can be beneficial, and our "XL" model was designed to explore the scalability of our approach in such settings.
> >
> > However, we agree that efficiency is crucial. As our model scaling analysis in **Table 4** shows, our architecture is flexible and not limited to large-scale configurations. Our smaller variants, such as the **0.63B "S" model**, also demonstrate competitive performance while being more resource-efficient. In addition, since SiTok operates at 12.5 Hz for both the encoder and decoder, it substantially reduces the sequence length, resulting in much lower memory consumption compared with transformer-based tokenizers that operate at 25–50 Hz. And, during inference for downstream language modeling, we don't need the decoder/encoder for understanding/generation.
> > In future work, we will further explore how SiTok can be adapted to settings with even more limited computational resources.
> >
> > ## Question
> >
> > ```Question 1: About the demo page.```
> >
> > Thank you for the question. We have prepared a demo page with audio samples for both direct **reconstruction and downstream TTS tasks** to better showcase our model's capabilities.
> >
> > To comply with the double-blind review policy, the demo page is hosted anonymously at the following link: [https://sitok-demo.github.io/](https://sitok-demo.github.io/).
> >
> > In addition, we are committed to open science and will **release the official code and pre-trained model checkpoints** upon the decision on our paper.
> >
> > ```Question 2: Ablation study on CTC loss supervision.```
> >
> > An excellent and insightful question! Your question correctly points out that a trade-off between semantic and acoustic fidelity is often expected. However, we believe a different dynamic is at play in our extreme low-bitrate setting.
> >
> > At a token rate of only 12.5 Hz, the information bottleneck is severe. Without semantic guidance, the reconstruction task is ill-posed. The model struggles to learn a meaningful representation and often collapses to producing an *"average"* acoustically ambiguous output that is poor in both content (high WER) and speaker identity (low SIM).
> >
> > The CTC loss acts as a powerful *structural regularizer*. By forcing the discrete tokens to align with linguistic units, it provides a robust *"semantic scaffold"* for the latent space. This has a crucial side effect: once the semantic content is anchored, the model can dedicate its remaining, limited representational capacity more efficiently to capturing the residual acoustic information, including speaker-specific characteristics.
> >
> > In essence, rather than creating a trade-off, the CTC supervision helps to disentangle semantic and acoustic features within the highly compressed latent space. It structures the optimization landscape, enabling the model to learn both aspects more effectively than it could with an unconstrained reconstruction objective alone. This is why we observe a marked improvement in both intelligibility (WER) and speaker similarity (SIM). We will elaborate on this insight in our ablation study discussion.

---

> > > ### Author Response · Authors · 2025-11-24
> > > **Response to reviewer xL4L (Part 3/3)**
> > >
> > > ```Question 3: Comparison with Tokenizers Trained for Understanding Tasks.```
> > >
> > > We thank the reviewer for this thoughtful suggestion to compare SiTok with tokenizers specifically optimized for understanding tasks. This is indeed an important point.
> > >
> > > While our primary goal is to develop a **universal tokenizer that balances both reconstruction and downstream speech lanuage modeling**, comparing against understanding-focused tokenizers provides a valuable benchmark for our model's representation learning capabilities.
> > >
> > > In fact, we have already included a relevant comparison in our original submission. The **GLM-4-Voice tokenizer**, included in Table 2, follows a design philosophy similar to S3Tokenizer (quantizing features from the ASR model Whisper) and operates at the same 12.5 Hz rate as our model. Our results show that SiTok outperforms it on all understanding tasks.
> > >
> > > To further address the reviewer's concerns, we have conducted additional experiments comparing SiTok with the **S3Tokenizer** used in CosyVoice. It is worth noting that CosyVoice's tokenizer operates at 25 Hz, giving it twice the temporal resolution of our 12.5 Hz model. Despite this disadvantage, SiTok still demonstrates superior or highly competitive performance, as shown below:
> > >
> > > | Model | FPS/TPS | Reconstruction | WER (↓) | SIM (↑) | ASR (↓) | ER (↑) | SV (↓) | KS (↑) |
> > > | :--- | :---: | :---: | :---: | :---: | :---: | :---: | :---: | :---: |
> > > | GLM4-Voice (S3-like) | 12.5/12.5 | Additional Token2Mel | 3.54 | below 0.2 (fix timbre) | 16.3 | 57.9 | 22.4 | 96.5 |
> > > | CosyVoice (S3) | 25/25 | Additional Token2Mel | 5.63 | 0.465 |10.02 | 61.2 | 16.6 | 97.1 |
> > > | **SiTok (CN=1)** | **12.5/12.5** | **Directly** | **4.06** | **0.641** | **4.95** | **63.5** | **13.8** | **96.9** |
> > >
> > > These results compellingly demonstrate that SiTok learns exceptionally strong representations for understanding tasks, even when compared to specialized tokenizers operating at higher frame rates. This is achieved without sacrificing the crucial ability for high-fidelity audio reconstruction, which underscores the effectiveness and versatility of our proposed approach.
> > >
> > > ---
> > >
> > > Thanks again for your constructive comments! We would be grateful if we could hear your feedback regarding our answers to the reviews. We would be happy to answer and discuss if you have further comments.
> > >
> > > ---
> > >
> > > References:
> > >
> > > [1] Du, Zhihao, et al. "Cosyvoice 2: Scalable streaming speech synthesis with large language models." arXiv preprint arXiv:2412.10117 (2024).
> > >
> > > [2] Wang, Xinsheng, et al. "Spark-tts: An efficient llm-based text-to-speech model with single-stream decoupled speech tokens." arXiv preprint arXiv:2503.01710 (2025).
> > >
> > > [3] Ye, Zhen, et al. "Llasa: Scaling train-time and inference-time compute for llama-based speech synthesis." arXiv preprint arXiv:2502.04128 (2025).

---

### Author Response · Authors · 2025-11-24
**General response to all reviewers (Part 1/2)**

First of all, we want to thank all the reviewers for your careful reading and for providing many constructive comments again. We also thank the ACs, SACs, and PCs for your coordination. In this overall response, we summarize several common issues raised across multiple reviews, explain how we address them, and indicate where these changes appear in the revised version of the paper. All newly added content will be highlighted in **blue** in the updated manuscript.

## 1. Zero-shot TTS Performance

Both reviewer **XL4L (weakness 1)** and **WeZq (question 1)** mentioned SiTok's performance in zero-shot TTS and acknowledged its strong reconstruction and downstream understanding performance. We agree that validating TTS performance is important, as it can further demonstrates that SiTok can be effective and efficient for speech language modeling.

During the rebuttal period, we added a **zero-shot TTS experiment**. Following standard practice, we trained a **0.5B language model** to autoregressively predict **SiTok speech tokens from text**. We trained this model on the **Emilia dataset** to better compare with the baselines. We refer to this system as *SiTok-AR-TTS*.

On the **SeedTTS test-en** set, SiTok-AR-TTS achieves: **WER: 2.46, SIM: 0.64, RTF: 0.234.**

More details about the TTS results can be found in our responses to reviewer XL4L (weakness 1) and WeZq (question 1). TTS samples are also available on our demo page. All TTS results and experimental details are included in the **Appendix C.2 (from page 17 to 18)** of the revised manuscript.

## 2. Demo Page

Both reviewer **XL4L (question 1)** and reviewer **5u8g (weakness 1)** suggested adding a demo page or providing reconstruction samples. We have added an anonymous demo page: [https://sitok-demo.github.io/](https://sitok-demo.github.io/).

It contains reconstruction results compared with baseline models as well as zero-shot TTS samples. We selected some in-the-wild and expressive speech examples to better showcase the reconstruction quality. Subjective evaluation with statistical significance tests is provided in the **Appendix C.3 (from page 18 to 19)** of the revised manuscript to complement the objective metrics.

## 3. Open-sourcing

Reviewer **5u8g (question 1)** asked whether the model will be open-sourced, and several reviewers requested more details about the architecture and training procedure.

We **confirm that we will open-source both the code and the pretrained model checkpoints (on public, research-only datasets)** after the paper decision, to further support reproducibility.

In addition, we have added the following materials to the **Appendix D Reproducibility Statement** of the revised manuscript:
- Detailed architectural specifications and pseudo-code for the encoder, codebook, diffusion decoder, and CTC regularizer (Appendix D.1, page 19 to 21),
- Pseudo-code outlining the core training loop for the end-to-end diffusion autoencoder (Appendix D.2, page 22),
- Training hyperparameters and data-processing steps (Appendix D.3, page 22 to 23).

These additions should make it significantly easier for others to reproduce our results or extend the framework.

## 4. Real-time Challenges

Reviewers **XL4L (weakness 3)**, **mPsW (question 1)**, and **WeZq (weakness 3)** all mentioned the lack of explicit real-time metrics such as RTF for speech reconstruction, since diffusion model inference is traditionally slow.

We appreciate the reviewers' positive feedback on our exploration of accelerating diffusion-based tokenizers through shortcut modeling. We have now added RTF results for both reconstruction and downstream TTS. Specifically, for reconstruction, SiTok achieves:

**RTF: 0.041 / 0.024 / 0.013** for **16 / 8 / 4** diffusion steps, respectively.

More discussion can be found in our responses to the corresponding reviewer questions. These results are also added to the **Section 3.3.5 Efficient Decoding (Page 7)** of the revised manuscript.

---

> ### Author Response · Authors · 2025-11-24
> **General response to all reviewers (Part 2/2)**
>
> ## 5. Novelty and Core Contribution
>
> Some reviewers raised concerns about novelty. Here, we briefly restate our core contribution and clarify our distinctions from prior work.
>
> In this work, our main motivation is to provide a **scalable paradigm** for training a tokenizer that simultaneously achieves high-quality reconstruction and strong downstream speech language modeling performance, at an **extremely low token rate (12.5 Hz)** and **200 bps bitrate** for efficient speech language modeling.
>
> We adopt an **end-to-end diffusion autoencoder** with **semantic regularization via a CTC decoder**, which jointly learns semantic-rich representations through supervised learning while enabling high-fidelity audio reconstruction through diffusion. This differs from prior two-stage diffusion tokenizers, which first train the encoder and quantizer and then train a diffusion decoder on fixed latents. Our experiments show that such end-to-end joint optimization is essential, compared with the two-stage alternative, for achieving both strong reconstruction quality and robust downstream performance. In addition, our extensive ablation studies provide further insights into the key design choices that contribute to SiTok’s overall effectiveness.
>
> ---
>
> ## All modifications in the revised version of the paper:
>
> - the demo page link in Abstract (page 1, line 19 to 20),
> - some discussion about other tokenizers with ASR-based loss (Section 2.2, page 3, line 143 to 146),
> - WER/SIM/UTMOS for grouth truth speech (Table 1, page 6, line 275),
> - RTF results for SiTok reconstruction (Section 3.3.5, page 7, line 360 to 364),
> - More details about the comparison between end-to-end diffusion and two-stage modeling (Section 3.4, page 8, line 399 to 400),
> - More discussion about related works (Appendix B, page 6),
> - Zero-shot TTS experiment and results (Appendix C.2, page 17 to 18),
> - Subjective evaluation (Appendix C.3, page 18 to 19),
> - Detailed architectural specifications and pseudo-code for SiTok model (Appendix D.1, page 19 to 21),
> - Pseudo-code outlining the core training loop for the end-to-end diffusion autoencoder (Appendix D.2, page 22),
> - Training hyperparameters and data-processing steps (Appendix D.3, page 22 to 23).

---

### Meta-Review · Area_Chair_enzw · 2026-01-04

**Summary:**

This work presents an algorithm for improving speech / audio tokenization using diffusion autoencoder / decoders, together with CTC and VQ. The reviewers agree that the paper presents an interesting alternative to tokenization, achieving strong compression (12.5 Hz) and good quality.

Issues that were raised during the review:
1. Missing TTS evaluations / relevant metrics.
2. Lack of clarity in presentation / missing details. [decoder / codebook configuration]
3. Real-time / streaming challenges.
4. Missing comparisons: With tokenizers trained for understanding tasks, waveform-based models, FSQ.
5. Concerns on novelty since diffusion based tokenziers and optimization with ASR-based supervision have been proposed in the past.
6. Issues with reproducibility due to the use of in-hours data.

Minor issues
1. Model size, restricting deployment on resource-constrained devices.
2. Missing demos.
3. Why does CTC loss help with similarity metric?

Due to concerns on novelty and streaming challenges, which the authors only partly address, the paper is truly borderline. But the authors did a good job providing additional details in their rebuttal.

**Reviewer Concerns:**

1. The authors argue that some prior works report reconstruction quality and semantic effectiveness, but that’s not a sufficient argument. One of the main goals of tokenization is the ability to generate high quality audio. So TTS evaluations should be included. The authors did include WER / SIM metrics in their rebuttal for pure TTS. But objective / subjective metrics like VISQOL, MUSHRA or MOS, which are typically used, are not included (the authors reported UTMOS, but VISQOL may be preferable since we have the reference). The authors also added CMOS in the appendix. This would have partially addressed the concerns, if not all.
2. The authors provide some clarifications, which likely would have addressed the concerns.
3. Streamability is a major shortcoming, and fast RT is not sufficient for streaming.
4. Added additional results that show competitive results compared to S3Tokenizer. The authors provide reasonable arguments as to why they chose mel-spectral representation + vocoder, and that some of the core ideas of the work can be used in waveform-based models (future work). They also provide results with FSQ.
5. Novelty: The authors clarify the placement of the current work. High compressions (12.5 Hz) and end-to-end optimization. The novelty here is marginal compared to prior work, in my opinion: decoder fine-tuning provides gains, which is similar to two-stage; so training the current model in two-stage might also work well / better (in the ablations, is unclear why does R perform worse than D in Understanding tasks since reconstruction / diffusion should ideally not affect understanding performance, assuming that’s the only change?).
6. Authors said they release checkpoints, code (not the data).

Minor.
1. Include results with smaller models.
2. Included a demo page. My assessment: Reconstruction sounds good. But zero shot has some minor issues with speaker similarity. But overall, the results sound pretty good.
3. The authors provide a hand-wavy argument. Would have been helpful to support it with some quantitative analysis.

**Reviewer Scores:**

Reviewer xL4L (Lei Xie): 4 -> 4/6

Reviewer mPsW (Md Fahim): 4 -> 4

Reviewer 5u8g (Xize Cheng): 6 -> 6

Reviewer WeZq (Xie Chen): 4 -> 4

Reviewer z58S (Avihu Dekel): 6 -> 6

---

### Decision · Program_Chairs · 2026-01-26

Accept (Poster)